# Evaluating the impact of spatial resolution on tropospheric NO₂ column comparisons within urban areas using high-resolution airborne data

Laura M. Judd[1,2], Jassim A. Al-Saadi[1], Scott J. Janz[3], Matthew G. Kowalewski[3,4], R. Bradley Pierce[5], James J. Szykman[6], Lukas C. Valin[6], Robert Swap[3], Alexander Cede[7], Moritz Mueller[7,8], Martin Tiefengraber[7,8], Nader Abuhassan[3,9], David Williams[6]

[1]NASA Langley Research Center, Hampton, VA, 23681, United States
[2]NASA Postdoctoral Program, Hampton, VA, 23681, United States
[3]NASA Goddard Space Flight Center, Greenbelt, MD, 20771, United States
[4]Universities Space Research Association, Columbia, MD, 21046, United States
[5]University of Wisconsin-Madison Space Science and Engineering Center, Madison, WI, 53706, United States
[6]United States Environmental Protection Agency Office of Research and Development, Triangle Research Park, NC, 27709, United States
[7]LuftBlick, Kreith, Austria
[8]Department of Atmospheric and Cryospheric Science, University of Innsbruck, Innsbruck, Austria
[9]Joint Center for Earth Systems Technology, University of Maryland-Baltimore County, Baltimore, MD, 21228, United States

*Correspondence to*: Laura M. Judd (laura.m.judd@nasa.gov)

**Abstract.** NASA deployed the GeoTASO airborne UV-Visible spectrometer in May-June 2017 to produce high resolution (approximately $250 \times 250$ m) gapless NO₂ datasets over the western shore of Lake Michigan and over the Los Angeles Basin. The results collected show that the airborne tropospheric vertical column retrievals compare well with ground-based Pandora spectrometer column NO₂ observations ($r^2$=0.91 and slope of 1.03). Apparent disagreements between the two measurements can be sensitive to the coincidence criteria and are often associated with large local variability, including rapid temporal changes and spatial heterogeneity that may be observed differently by the sunward viewing Pandora observations. The gapless mapping strategy executed during the 2017 GeoTASO flights provides data suitable for averaging to coarser areal resolutions to simulate satellite retrievals. As simulated satellite pixel area increases to values typical of TEMPO, TROPOMI, and OMI, the agreement with Pandora measurements degraded, particularly for the most polluted columns as localized large pollution enhancements observed by Pandora and GeoTASO are spatially averaged with nearby less-polluted locations within the larger area representative of the satellite spatial resolutions (aircraft-to-Pandora slope: TEMPO scale=0.88; TROPOMI scale=0.77; OMI scale=0.57). In these two regions, Pandora and TEMPO or TROPOMI have the potential to compare well at least up to pollution scales of $30 \times 10^{15}$ molecules cm⁻². Two publicly available OMI tropospheric NO₂ retrievals are both found to be biased low with respect to these Pandora observations. However, the agreement improves when higher resolution a priori inputs are used for the tropospheric air mass factor calculation (NASA V3 Standard Product slope = 0.18 and Berkeley High Resolution Product slope=0.30). Overall, this work explores best practices for satellite validation strategies with Pandora direct-sun observations by showing the sensitivity to product spatial resolution and demonstrating how the high spatial resolution NO₂ data retrieved from airborne spectrometers, such as GeoTASO, can be used with high temporal resolution ground-based column observations to evaluate the influence of spatial heterogeneity on validation results.

## 1 Introduction

Nitrogen oxides (NOₓ: NO + NO₂) are primarily emitted via fossil fuel combustion, soil microbial processes, biomass burning and lightning. NO₂ is an important precursor of ozone and particulate matter, making it one of the six criteria air

pollutants monitored by the United States Environmental Protection Agency (EPA: https://www.epa.gov/criteria-air-pollutants).  Unlike less reactive trace gases having atmospheric lifetimes of days or longer, the atmospheric lifetime of $NO_x$ is reported to be on the order of hours in the daytime polluted boundary layer (Liang et al. 1998; Beirle et al., 2011; Liu et al., 2016).  This short lifetime along with large variations in emission rates from sources causes the spatial distribution of $NO_2$ in polluted regions to be highly heterogeneous, making it difficult to characterize over urban areas without high spatiotemporal observations.

Since the 1990s, $NO_2$ column densities have been monitored globally from sun-synchronous satellite platforms utilizing the differential optical absorption spectroscopy (DOAS) methodology applied to earth-shine spectra in the visible-blue wavelengths.  The nadir spatial resolution of these sensors has generally improved over time—beginning with the Global Ozone Monitoring Experiment (GOME) at 40 x 320 km in 1995 (Burrows et al., 1999), dramatically refining a decade later with the Ozone Monitoring Instrument (OMI) at 13 x 24 km in 2004 (Levelt et al., 2006), and improving most recently to a sub-city spatial scale from the TROPOspheric Monitoring Instrument (TROPOMI) instrument aboard Sentinel-5P launched October 2017 with a nadir spatial resolution of 3.5 x 7 km (van Geffen et al., 2019).

OMI has been a prominent resource for understanding the global distribution of tropospheric $NO_2$ since its launch.  However, short-falls have been documented regarding its inability to capture the spatial variability within polluted regions (e.g., Valin et al., 2011a, Valin et al., 2011b; Broccardo et al., 2018). This inability is further hindered by the use of coarse a priori assumptions in the air mass factor (AMF) calculation for slant-to-vertical column conversion (Heckel et al., 2011; Russell et al., 2011; Goldberg et al., 2017). Though TROPOMI observes $NO_2$ at a spatial resolution an order of magnitude finer than OMI, its sun-synchronous orbit still limits lower- to mid-latitude observations to the early afternoon hours.

However, within the next decade three geostationary air quality monitoring missions will be launched that will be able to monitor daytime air quality hourly at spatial scales of less than 10 km. These missions include observations of North America with Tropospheric Emissions: Monitoring Pollution (TEMPO) (Zoogman et al., 2017), Asia with Geostationary Environment Monitoring Spectrometer (GEMS) (Kim et al., 2017), and Europe with Sentinel-4/Ultraviolet/Visible/Near-Infrared Instrument (UVN) (Ingmann et al., 2012).  These geostationary measurements over industrialized regions of the Northern Hemisphere along with ongoing global daily sun-synchronous measurements will be important contributors to the goal of creating an atmospheric composition global observing network (IGACO, 2004; CEOS, 2011).

To prepare for these planned geostationary air quality missions, NASA supported the development of airborne UltraViolet-VISible (UV-VIS) mapping instruments, (Geostationary Trace Gas and Aerosol Sensor Optimization: GeoTASO and GEO-CAPE Airborne Simulator: GCAS), to help determine the satellite instrument requirements for measurements relevant to air quality and to facilitate retrieval algorithm development at fine spatial resolutions at all times of day (Leitch et al., 2014; Kowalewski and Janz, 2014; Nowlan et al., 2016; Lamsal et al., 2017; Nowlan et al., 2018). These instruments have the capability of retrieving $NO_2$ at sub-kilometer spatial resolutions which can be useful in assessing trace gas heterogeneity at spatial scales finer than those of current and planned space-based retrievals. Similar instruments have also been developed by other countries. In the last decade, airborne spectrometers have mapped high resolution $NO_2$ in parts of Europe (Popp et al., 2012; Schönhardt et al 2015; Lawrence et al., 2015; Meier et al., 2017; Tack et al., 2017; Tack et al 2019), the United States (Nowlan et al., 2016; Lamsal et al., 2017; Nowlan et al., 2018; Judd et al., 2018), Asia (Judd et al., 2018), and Africa (Broccardo et al., 2018).  Results from these efforts have illustrated the capability of airborne spectrometers to observe the impact of emission sources and meteorology on the spatial distribution of $NO_2$ (Popp et al., 2012, Schönhardt et al., 2015; Judd et al., 2018, and Tack et al., 2019) and have shown utility for evaluating NOx emissions (Schönhardt et al 2015; Souri et al 2018) as these high resolution measurements can resolve detailed $NO_2$ spatial patterns that satellites, at their current spatial resolutions, cannot (Broccardo et al., 2018; Lamsal et al., 2017; Judd et al., 2018).

Measurements from these airborne UV-VIS mapping instruments can provide a transfer standard between the space-based sensor footprint and ground-based column measurements used for validation of the satellite trace gas products, such as $NO_2$ troposperic vertical columns. The direct-sun DOAS technique used to retrieve $NO_2$ from ground-based Pandora spectrometer measurements has been shown to be highly precise and accurate due to little uncertainty in the path light travels through the atmosphere (i.e., the air mass factor) at solar zenith angles less than 80 degrees (Herman et al., 2009). Thus, Pandora instruments that operate in direct-sun mode are a strong candidate for providing a validation standard for trace gas retrievals from geostationary sensors like TEMPO as well as low Earth orbiting sensors like TROPOMI. NASA and EPA have been working toward creating long-term measurement sites across the United States to prepare for and aid validation of the satellite products and their usage in air quality management activities (EPA, 2019a). Developments of

similar validation site activities are also underway in East Asia and Europe. However, in heterogeneously polluted areas such as cities, the relatively local measurements at such sites may not necessarily be representative of the spatial scales observed by space-based platforms. High resolution airborne mapping observations provide a unique perspective to assess the impact of spatial scale mismatches between satellite observations and local validation site measurements.

This work compares high spatial resolution $NO_2$ vertical columns retrieved from GeoTASO to those measured from small networks of Pandora spectrometers operating in direct-sun mode installed in two regions: the western shore of Lake Michigan and the Los Angeles (LA) Basin. The high spatial resolution GeoTASO data are then upscaled to spatial resolutions typical of past, present, and planned space-based sensors to demonstrate sensitivity of these comparisons to satellite pixel size in an idealized framework. Finally, these results are compared to two publicly available OMI $NO_2$

retrievals to provide real-world context. Overall, this work shows the sensitivity of satellite product validation strategies to spatial resolution and begins to demonstrate how the high spatial resolution $NO_2$ data retrieved from airborne mapping observations can be used with planned high temporal resolution ground-based column observations to evaluate the influence of spatiotemporal heterogeneity on satellite-based trace gas product validation.

## 2 Data

### 2.1 Campaign Overview

    In 2017, one of NASA's airborne UV-VIS mapping instruments, GeoTASO, was flown aboard the NASA Langley Research Center UC-12b aircraft as part of the Lake Michigan Ozone Study (LMOS; https://www-air.larc.nasa.gov/missions/lmos/index.html) and during the Student Airborne Research Program (SARP; https://airbornescience.nasa.gov/content/Student_Airborne_Research_Program). Table 1 summarizes all GeoTASO flights

that occurred from May 22nd-June 27th, 2017. GeoTASO during LMOS was used to characterize the influence of $NO_2$ emissions and transport on ozone exceedances along the western shore of Lake Michigan. During LMOS, GeoTASO flew on 21 research fights totaling over 90 hours. The left map in Figure 1 shows the areas mapped by GeoTASO during the month-long LMOS campaign in grey. The entire LMOS domain cannot be mapped during a single flight, therefore individual flight plans focused on sub-portions of the domain where air quality and meteorological forecasts suggested that science goals

could be met for each flight. Eight of the research flights focused on the city of Chicago, whereas the rest were north of the city along the western shore of Lake Michigan. Each flight lasted approximately four hours and up to two flights were flown per day (one morning and one afternoon). Flights were optimized for times where clear skies were expected and any cloudy scenes that resulted are filtered out for this analysis. The median solar zenith angle for all flights was 36° with a minimum/maximum of 19°/70°.

After LMOS, five research flights were conducted on June 26th and 27th, 2017, totaling 15 hours, in the Los Angeles (LA) Basin during SARP. In 2017, the LA Basin had the highest 8 h and 1 h ozone concentrations in the United States, as well as the highest annual mean of $NO_2$ in the United States (EPA, 2019b). In the LA Basin, a single flight plan was developed to maximize coincidences with Pandora spectrometers installed in the region and extended south to the industrial area near Long Beach, CA (Fig. 1 right). This plan was completed once per flight with up to three flights per day

(each flight lasting approximately three hours). Both flight days in the LA Basin were cloud-free. LA Basin flights spanned a larger range of solar zenith angles than LMOS with a median of 49° and a minimum/maximum of 10°/70°.

    One Cimel sunphotometer was operating in both study domains during flight days (https://aeronet.gsfc.nasa.gov/cgi-bin/bamgomas_interactive). At the Zion site during LMOS, AOD was generally below 0.1 on flight days with a peak reported of ~0.2 after cloud filtering. In the LA Basin, there were no data reported for the June

26th, 2017 flight day, but AODs were less than 0.1 on June 27th, 2017 and both days were similar meteorologically.

    The LMOS region and the LA Basin each had five Pandora instruments operating in direct-sun mode within the research areas during GeoTASO flight days (hexagons in Fig. 1). The number of valid coincidences for each flight are labeled in Table 1. All flight plans had planned overflights of multiple Pandora sites, but cloud cover or instrument issues may have resulted in a fewer (or zero) valid coincidences. Pandora site names and coordinates and observed pollution

ranges during coincidences are listed in Table 2. The locations of these instruments span a wide variety of pollution environments. Urban locations include all those located in the LA Basin and one near the Chicago O'Hare Airport (Schiller

Park). Pandora spectrometers located along the western shore of Lake Michigan north of Chicago during LMOS were the cleanest (Zion, Milwaukee, Grafton, Sheboygan).

**Table 1: GeoTASO flight summary**

| Flight | Date | Time | Location | SZA Range (°) | Pollution Scale (95th percentile x10$^{15}$ molecules cm$^{-2}$) | Percent Cloudy Pixels | # Valid Pandora Coincidences |
|---|---|---|---|---|---|---|---|
| 1 | 2017-05-22 | 12.4-16.3 UTC | Chicago, IL | 29-70 | 12.6 | 1% | 3 |
| 2 | 2017-05-22 | 17.7-21.2 UTC | Chicago, IL | 21-47 | 3.6 | 29% | 1 |
| 3 | 2017-05-27 | 16.2-19.9 UTC | Zion-to-Sheboygan | 20-33 | 2.4 | 3% | 7 |
| 4 | 2017-06-01 | 13.3-17.1 UTC | Chicago, IL | 22-59 | 17.1 | 0% | 2 |
| 5 | 2017-06-01 | 19.1-22.8 UTC | Zion-to-Sheboygan | 26-64 | 6.8 | 0% | 7 |
| 6 | 2017-06-02 | 19.1-23.0 UTC | Zion-to-Sheboygan | 26-66 | 9.5 | 0% | 4 |
| 7 | 2017-06-04 | 19.3-22.6 UTC | Chicago, IL | 26-64 | 10.1 | 7% | 3 |
| 8 | 2017-06-07 | 12.9-16.9 UTC | Zion Area | 23-63 | 3.4 | 0% | 4 |
| 9 | 2017-06-07 | 18.7-22.7 UTC | Zion Area | 23-62 | 6.0 | 0% | 3 |
| 10 | 2017-06-08 | 12.8-16.6 UTC | Zion Area | 26-64 | 5.0 | 0% | 3 |
| 11 | 2017-06-12 | 18.9-22.5 UTC | Zion Area | 24-59 | 6.0 | 7% | 1 |
| 12 | 2017-06-13 | 16.4-19.7 UTC | Chicago, IL | 18-29 | 5.5 | 9% | 1 |
| 13 | 2017-06-14 | 13.3-17.0 UTC | Chicago, IL | 21-59 | 5.9 | 8% | 3 |
| 14 | 2017-06-15 | 18.8-22.7 UTC | Sheboygan Area | 22-47 | 2.1 | 3% | 2 |
| 15 | 2017-06-15 | 18.7-22.7 UTC | Sheboygan Area | 22-61 | 5.2 | 2% | 9 |
| 16 | 2017-06-17 | 16.3-20.3 UTC | Sheboygan Area | 20-36 | 0.1 | 24% | 1 |
| 17 | 2017-06-18 | 13.3-17.3 UTC | Chicago, IL | 21-60 | 3.2 | 14% | 2 |
| 18 | 2017-06-19 | 13.4-17.1 UTC | Chicago, IL | 22-58 | 8.3 | 25% | 1 |
| 19 | 2017-06-21 | 13.2-17.1 UTC | Zion-to-Sheboygan | 22-60 | 7.1 | 2% | 1 |
| 20 | 2017-06-21 | 19.0-22.0 UTC | Zion-to-Sheboygan | 24-53 | 5.7 | 16% | 0 |
| 21 | 2017-06-26 | 15.1-17.4 UTC | LA Basin | 35-62 | 25.7 | 0% | 5 |
| 22 | 2017-06-26 | 22.7-1.5 UTC | LA Basin | 38-70 | 17.3 | 0% | 4 |
| 23 | 2017-06-27 | 14.7-17.2 UTC | LA Basin | 37-67 | 29.8 | 0% | 5 |
| 24 | 2017-06-27 | 18.7-21.1 UTC | LA Basin | 10-20 | 17.0 | 0% | 3 |
| 25 | 2017-06-27 | 22.7-1.2 UTC | LA Basin | 38-70 | 22.0 | 0% | 5 |

# LMOS

# LA Basin

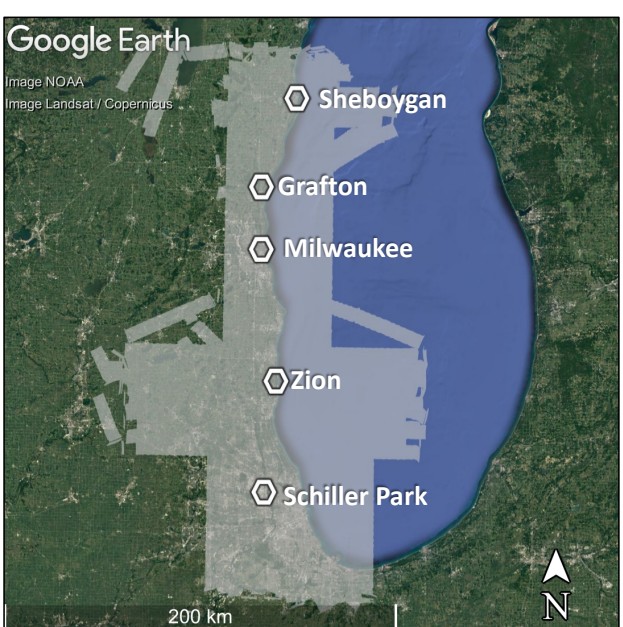
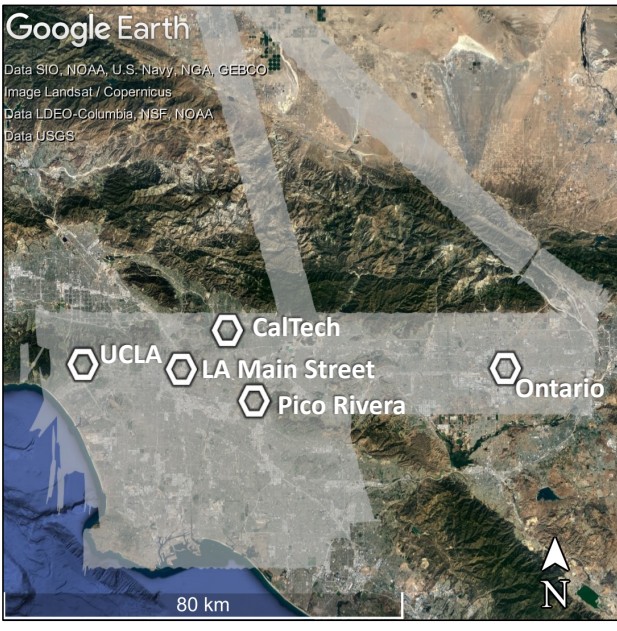

**Figure 1: Maps of the LMOS and LA Basin regions showing the areas mapped by GeoTASO during summer 2017. The labeled hexagons indicate locations of Pandora spectrometers operating during the flight days.**

5    **Table 2: Pandora site names, location, and TropVC Range during GeoTASO coincidences in the LMOS and LA Basin domains. These locations are also indicated by hexagons on the maps in Figure 1.**

| Pandora Site | Latitude, Longitude (°) | TropVC Range during GeoTASO Coincidences x$10^{15}$ molecules cm$^{-2}$ |
|---|---|---|
| LA Main Street | 34.066, -118.227 | 7.4-28.5 |
| Ontario | 34.068, -117.526 | 7.5-30.1 |
| UCLA | 34.074, -118.441 | 5.0-24.0 |
| Pico Rivera | 34.010, -118.069 | 12.1-18.3 |
| CalTech | 34.136, -118.127 | 3.4-14.2 |
| Sheboygan | 43.746, -87.709 | -0.9-4.3 |
| Grafton | 43.343, -87.920 | -0.2-4.0 |
| Zion | 42.468, -87.810 | 1.5-8.0 |
| Schiller Park | 41.965, -87.876 | 3.5-45.1 |
| Milwaukee | 43.061, -87.914 | 4.1-5.9 |

**2.2 Pandora**

Pandora is a UV-VIS ground-based spectrometer used to retrieve trace gas column amounts (Herman et al., 2009; Herman et al., 2015). Although the instrument has the capability to make both direct-sun and all-sky radiance measurements from which trace gas amounts can be retrieved, only direct-sun retrievals were collected during these campaigns. Direct-sun $NO_2$ columns are reported to have high precision and accuracy and they are availability in near-real time as a standard product. The direct-solar beam is measured by Pandora via an optical head sensor attached to a solar tracker. The solar beam is carried to the UV-VIS spectrometer by a fiber optic cable attached to the head sensor. $NO_2$ is retrieved by applying a spectral fitting algorithm (Cede, 2017) using a near-noon reference spectrum, from which the $NO_2$ slant column amount is derived by a statistical calibration approach (Herman et al., 2009). A geometric AMF is used for the slant-to-vertical column conversion as the path length through the atmosphere is dominated by the direct solar beam rather than scattered light through the atmosphere (Cede, 2017). The precision and accuracy of the direct sun $NO_2$ vertical columns are reported as $2.67 \times 10^{14}$ and $2.67 \times 10^{15}$/AMF molecules $cm^{-2}$, respectively, by Herman et al. (2009). In this work the Pandora retrievals are screened to exclude observations with vertical column error greater than 0.05 DU and normalized RMS greater than 0.005, to limit the retrieval uncertainty to approximately 10%.

**2.3 Ozone Monitoring Instrument (OMI)**

OMI is a space-based UV-VIS instrument launched in 2004 aboard the EOS-Aura satellite in a sun synchronous orbit with an equator crossing time in the early afternoon (Levelt et al., 2006). This work uses the NASA Standard Product (SP) Version 3 (Krotkov et al., 2017) and the Berkeley High Resolution (BEHR) Product (Russell et al., 2011; Laughner et al., 2018b; Laughner et al., 2019) to assess how their tropospheric column retrievals compare with measurements from Pandora spectrometers deployed during summer 2017 in the LMOS and the LA Basin domains.

The two vertical column products are based on the same slant columns (produced by the SP retrieval) but they differ in their a priori inputs for the tropospheric air mass factor calculation. As the name suggests, BEHR a priori have a higher spatial resolution than the NASA SP (Russell et al., 2011; Bucsela et al., 2013; Krotkov et al., 2017; Laughner et al., 2018a; Laughner et al., 2019). Both products use monthly $NO_2$ a priori profile assumptions, but NASA SP V3 profiles are at a spatial resolution of 1° latitude x 1.25° longitude from the Global Modeling Initiative chemical transport model whereas BEHR profiles are from a WRF-Chem model simulation at 12 km resolution. Assumptions about surface reflectivity in the NASA SP are from an OMI based climatology at 0.5° x 0.5° (Kleipool et al., 2008) and the BEHR product uses a bidirectional reflectance factor from the Moderate Resolution Imaging Spectroradiometer (MODIS) over land at 30 arcsec resolution and an ocean reflectivity model over water. Terrain pressure in BEHR is assumed from WRF-Chem pressure profiles adjusted with the Global Land One-kilometer Base Elevation database (Hastings and Dunbar, 1999) and the hypsometric equation whereas the NASA SP uses a 3 km digital elevation model (Boersma et al., 2011).

OMI data used in this analysis are filtered to exclude cloud fractions (derived from MODIS) greater than 20% (following criteria defined in Laughner et al. (2019)) and data from OMI's row anomaly (http://projects.knmi.nl/omi/research/product/rowanomaly-background.php). From 2004 until October 2017, OMI had the finest spatial resolution of any $NO_2$ retrieving sensor in low Earth orbit. Before the appearance of the row anomaly in 2007, the finest pixel size of OMI was approximately 13 x 24 km at nadir. In January 2009, the row anomaly extended to affect nadir pixels and during summer 2017 the pixel areas coincident with the campaign Pandora observations ranged from 365 $km^2$ near-nadir to approximately 4600 $km^2$ at the edge of the swath.

**2.4 GeoTASO**

Geostationary Trace gas and Aerosol Sensor Optimization (GeoTASO) is a hyperspectral mapping instrument built by Ball Aerospace (Leitch et al, 2014) to acquire data for optimizing and testing new high-resolution retrievals of trace gases ($NO_2$, $O_3$, and HCHO) and aerosols in preparation for geostationary air quality observations. The instrument is composed of a reflective telecentric telescope that focuses scattered light within its field of view through a photo-elastic modulator for depolarization and lastly into an Offner spectrometer. Within the spectrometer the second order and first order diffracted beams are used to form the ultraviolet (UV) and visible (VIS) spectral ranges, respectively. Each spectral range (UV: 300

nm to 380 nm and VIS: 410 nm to 690 nm) is focused onto a separate 2-dimensional charge coupled device (CCD) (1056 pixels in the wavelength dimension x 1033 pixels in the spatial dimension) operated at an integration time of 250 ms for all flights. $NO_2$ is retrieved using a spectral window within the VIS channel, which has a spectral resolution (FWHM) of 0.88 nm and a spectral sampling of 3.1 pixels (Nowlan et al., 2016).  GeoTASO observes as a push-broom sensor with an across-track nadir field of view of 45°, providing a swath-width of approximately 7 km from a nominal altitude of 8.5 km. Spectra (~300 images) are coadded to an approximate ground pixel size of 250 x 250 m.

During GeoTASO flights, gapless maps (otherwise referred to as rasters) were created over an area of interest by executing flight plans composed of parallel flight legs spaced 6 km apart. The overlap in swath between these flight legs provided a tolerance for maintaining gapless coverage during instances of strong cross-winds or when a lower flight altitude was selected in order to operate below high-level thin cirrus clouds.  This raster strategy provides data that can easily be spatially co-added to evaluate the influence of spatial resolution on tropospheric $NO_2$ column measurements, as is shown herein.  Additional details and estimated uncertainties of the GeoTASO $NO_2$ retrieval are elaborated in the following sections.

### 2.4.1 Airborne $NO_2$ Slant Column Retrieval and Uncertainty

$NO_2$ differential slant columns (DSCs) are retrieved using QDOAS, an open-source DOAS computing software developed by the Royal Belgian Institute for Space Aeronomy (http://uv-vis.aeronomie.be/software/QDOAS). Fitted trace gas and interference cross sections include $NO_2$ (Vandaele et al., 1998), $O_4$ (Thalman and Volkamer, 2013), $H_2O$ (Rothman et al., 2009), CHOCHO (Volkamer et al., 2005), Ring spectrum (Chance and Kurucz, 2010), and a fifth-order polynomial in the spectral window of 425-460 nm. The resultant DSCs can be physically described as the additional total $NO_2$ absorption along the path sunlight travels through the atmosphere to the GeoTASO instrument relative to the reference atmosphere.

Reference spectra are collected in-flight using nadir-viewing observations for each across track position in the swath over a homogeneous area with minimal $NO_2$ absorption. One reference set was collected for each region.  The LMOS reference spectrum was observed during nadir observations on May 27[th], 2017 at 19:00 UTC (SZA=35°) over Lake Michigan with an estimated below aircraft column of $2.3x10^{15}$ molecules cm$^{-2}$ (estimated from the NAM-CMAQ model described below).  For the LA Basin flights, the reference was collected over a clean homogeneous area north of the LA Basin on June 27[th], 2017 also at 19:00 UTC (SZA=16°) with an estimated below aircraft column of approximately $1x10^{15}$ molecules cm$^{-2}$.

The average spectral fitting uncertainty for the $NO_2$ DSCs over all flights is $1.2x10^{15}$ molecules cm$^{-2}$ with a standard deviation of $0.25x10^{15}$ molecules cm$^{-2}$. Higher solar zenith angles (SZA) and lower surface albedos both result in slightly higher uncertainties due to their impacts on instrument signal-to-noise ratio. A multi-linear regression applied to the dependent variable of $NO_2$ DSC uncertainty vs. the independent variables of SZA and surface albedo, for all cloud-free data, indicates that 33% of the variability in DSC uncertainty is associated with changes in SZA (uncertainty increases with increasing SZA by $0.011x10^{15}$ molecules cm$^{-2}$ per degree) and almost 15% is associated with variations in surface albedo (uncertainty increases with decreasing albedo by $0.033x10^{15}$ molecules cm$^{-2}$ per 0.01 decrease in albedo). Additional uncertainty due to using one $NO_2$ absorption cross section at a single temperature (294 K) leads to a potential bias of -0.6±1.7%, estimated using the $NO_2$ profile weighted effective temperature profile in Eq. 4 in Bucsela et al. (2013).

### 2.4.2 Slant-to-Vertical Column Conversion

Further processing of DSCs into vertical columns (VCs) requires computations of the AMF: the ratio of the pathlength of light through the atmosphere due to scattering and geometry to the vertical pathlength.  AMFs depend on a priori assumptions such as the vertical distribution of $NO_2$, surface reflectivity, clouds, aerosols, sun angle, and viewing geometry (Palmer et al., 2001; Lamsal et al., 2017).  Mathematically, AMFs are the integrated product of (1) scattering weights (the vertical distribution of the instrument sensitivity calculated by a radiative transfer model) and (2) the $NO_2$ profile shape factor (the relative vertical distribution of $NO_2$) (Palmer et al., 2001). Scattering weight calculations are most sensitive to inputs that influence light transmission through the atmosphere, most notably from solar and viewing geometry, surface reflectivity, and aerosols (Lamsal et al. 2017; Meier et al., 2017).  This work uses Harvard Smithsonian Astrophysical Observatory's AMF Tool which packages the VLIDORT radiative transfer model to calculate scattering weights (Spurr,

2006; Nowlan et al., 2016; Nowlan et al., 2018).  Inputs include atmospheric profiles of temperature, pressure, ozone, $NO_2$, aerosol, aircraft altitude, viewing geometry, solar geometry, and surface reflectance.

Tropospheric $NO_2$ profiles are taken from a 12 km hourly analysis of a parallel developmental simulation of the North American Model-Community Multiscale Air Quality (NAM-CMAQ) model from the National Air Quality Forecasting Capability (NAQFC; Stajner et al., 2011).  Stratospheric profiles are estimated using the PRATMO chemical box model of stratospheric $NO_2$ profile climatology (Prather, 1992; McLinden et al. 2000) which estimates $NO_2$ between approximately 10 and 60 km as a function of month, latitude, and solar zenith angle.  This stratospheric model has been used in previous GeoTASO retrievals and has an estimated uncertainty of ~30% in the stratospheric column (Bourassa et al., 2011; Nowlan et al., 2016). Temperature and pressure profiles from the NAM-CMAQ in the troposphere and the Realtime Air Quality Modeling System (RAQMS; Pierce et al., 2009) up to 60 km are extracted and merged, and ozone profiles are extracted from the NAM-CMAQ analysis for the troposphere and OMI gridded monthly climatology in the stratosphere (Liu et al. 2010) for May and June 2017.

Noguchi et al. (2014) evaluated the influence of surface anisotropy on $NO_2$ AMFs for geostationary scale measurements using a 1 km MODIS BRDF product and found that not accounting for BRDF in the AMF calculation, especially in areas with high $NO_2$ near the surface, can lead to large errors.  For this reason, this work uses the BRDF isometric, volumetric, and geometric kernels retrieved in Band 3 by the MODIS MCD43A1 daily L3 500m v006 product (Lucht et al., 2000; Schaaf and Wang, 2015).  To fill gaps and decrease noise, the daily product is averaged over a month in the regions GeoTASO measured (May 22nd-June 21nd for LMOS and June 15th-July 14th for the LA Basin). Remaining gaps are small and are filled using linear interpolation of nearby pixels. Variations in BRDF derived albedo compare well with the brightness of surface features. Li et al. (2018) found the v006 MCD43A1 product to compare very well with Multi-angle Imaging SpectroRadiometer (MISR) land surface reflectance with little bias and a high correlation (r=0.95 in a case study in northeast Asia).

Clouds and aerosols are not considered in the AMF calculation in this work, as cloudy scenes are filtered by removing pixels having detector count rates of greater than $2.5 \times 10^4$ counts $s^{-1}$ within the DOAS $NO_2$ spectral window and aerosol loading was relatively low during flight days.  Given the relatively low aerosol loadings, errors in AMF due to neglecting aerosols are expected to be small for these flights (< 5% using sensitivities shown by Lamsal et al. (2017)).

Differential slant columns are converted into below aircraft vertical columns using the following equation:

$$VC_{below} \text{ or } TropVCs = \frac{DSC - VC_{above}AMF_{above} + reference\ slant\ column}{AMF_{below}}, \qquad (1)$$

Where DSC is the differential slant column, $VC_{above}$ and $VC_{below}$ are respectively the vertical column above and below the aircraft, and $AMF_{above}$ and $AMF_{below}$ are respectively the calculated AMF above and below the aircraft. As $NO_2$ column variations in the troposphere are largely confined to the boundary layer, this work assumes that residual tropospheric variations in the above-aircraft column are negligible and the below aircraft vertical columns are referred to as tropospheric vertical columns (TropVCs).

**2.4.3 Vertical Column Retrieval Sensitivity and Uncertainty**

Figure 2 shows examples of $NO_2$ TropVCs (top) and tropospheric AMFs (middle) for a morning flight (~08:30-12:00 LDT) over Chicago, IL on June 1st, 2017 (left) and a morning flight (~8:40-10:05 LDT) over the LA Basin from June 26th, 2017 (right).  The bottom row of Fig. 2 shows base maps from Google Earth Pro to aid identification of surface features in the maps above.  In the areas mapped in Fig. 2, $NO_2$ TropVC spans roughly two orders of magnitude and spatial patterns are consistent with emission sources, such as busy roadways, airports, and industrial regions. Both mornings in each region had relatively stagnant conditions.

**20170601 Morning Chicago Raster**     **20170626 Morning LA Basin Raster**

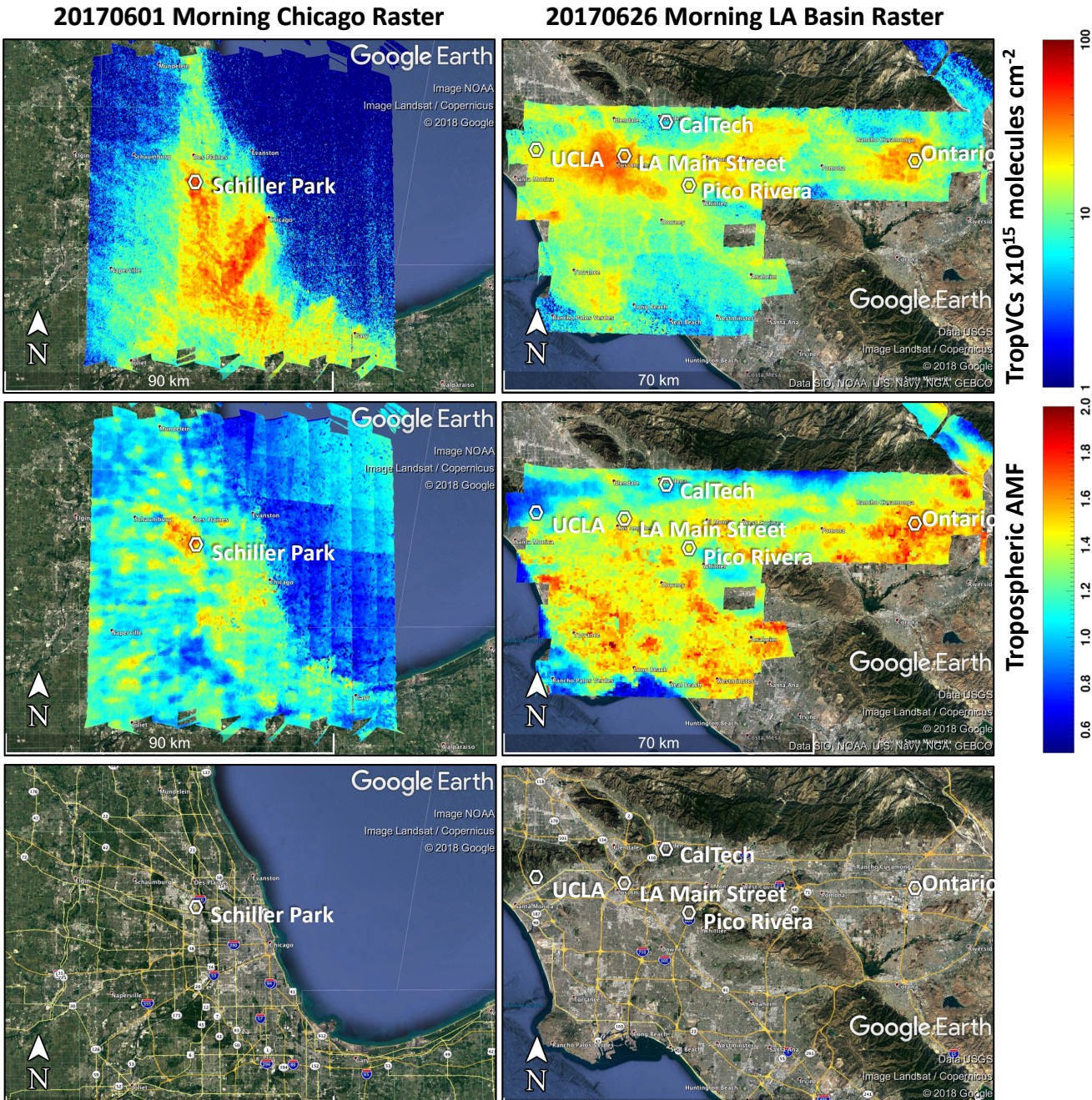

**Figure 2: Tropospheric NO₂ vertical column on a log10 color scale (top), tropospheric AMF on a linear color scale (middle), and base map (bottom) for a Chicago raster on the morning (08:30-12:00 LDT) of June 1st, 2017 (left) and an LA Basin raster on the morning (08:40-10:05 LDT) of June 26th, 2017 (right). The base maps show roadways, urbanized areas, and surface characteristics. The labeled hexagons indicate locations of Pandora spectrometers.**

Over land, large spatial variations of tropospheric AMF are evident and associated primarily with the surface reflectance characteristics. A multi-variable linear regression of these data indicates that surface albedo explains 64% of the variability in tropospheric AMF, illustrating the importance of accurately treating surface reflectance in these high spatial resolution retrievals. Over an area with homogeneous surface reflectance (like Lake Michigan east of Chicago), the subtle

influence of a varying $NO_2$ shape factor is visible in the AMF, but has little impact on the magnitude of TropVC because there is little $NO_2$ in this location. The a priori $NO_2$ profile accounts for 16% of the variability in tropospheric AMF. Viewing geometry has smaller effects, with SZA accounting for 1.8% of the variability and the relative azimuth angle and viewing zenith angle causing even smaller amounts.

To better gauge the sensitivity of AMF and TropVC to the $NO_2$ vertical profile in the troposphere, the AMFs

calculated for the most polluted LMOS flight (June 1st, 2017; Fig. 2) were compared to AMFs calculated using a single median NAM-CMAQ $NO_2$ vertical profile (not shown). This median profile is representative of a moderately polluted urban area with a tropospheric $NO_2$ column of $3.6 \times 10^{15}$ molecules $cm^{-2}$, most of the $NO_2$ in the lowest km of the atmosphere, and an $NO_2$ maximum at the surface of approximately 4 ppbv. The resulting differences in AMF can be large, ranging from -30% to 10%, however the largest AMF differences occur over relatively clean areas in which the median polluted profile shape is

less representative. The differences in TropVCs are about $\pm 1 \times 10^{15}$ molecules $cm^{-2}$ at the 5th/95th percentiles with 50% of the points having differences between $-0.2 \times 10^{15}$ and $0.5 \times 10^{15}$ molecules $cm^{-2}$. These values are on the same order of magnitude as the uncertainty in DSCs.

To estimate total uncertainty in the TropVC, error propagation was applied to Eq. (1). Assuming a 50% uncertainty in the NAM-CMAQ profile, 30% in the PRATMO profiles, and a conservative 30% uncertainty in the below aircraft AMF

(largest percent variation in the sensitivity study discussed in the preceding paragraph), the uncertainty in the reference slant column is estimated to be $2.3 \times 10^{15}$ molecules $cm^{-2}$ for LMOS and $1.9 \times 10^{15}$ molecules $cm^{-2}$ for SARP. The average uncertainty in the above aircraft vertical column is $1.3 \times 10^{15}$ molecules $cm^{-2}$ with a standard deviation of $0.08 \times 10^{15}$ molecules $cm^{-2}$. The median total uncertainty for TropVCs for both regions combined is $2.3 \times 10^{15}$ molecules $cm^{-2}$. Percent uncertainty (5th-95th percentile) ranges from 46-143% for relatively unpolluted columns ($< 5 \times 10^{15}$ molecules $cm^{-2}$), from 32-60% for

moderately polluted columns ($5-15 \times 10^{15}$ molecules $cm^{-2}$), and ~30% for polluted columns ($> 15 \times 10^{15}$ molecules $cm^{-2}$). These values are similar to those calculated by Nowlan et al. (2016).

## 3 Results

### 3.1 Comparison of airborne GeoTASO and ground-based Pandora retrievals

Pandora spectrometer measurements have been previously used to validate retrieval products from GeoTASO and similar

airborne mapping spectrometers during intensive campaigns (Nowlan et al., 2016; Lamsal et al., 2017; Nowlan et al., 2018). Figure 3 shows the comparison between the 80 coincident Pandora and GeoTASO TropVC observations during LMOS and the LA Basin flights colored by ground site. The first five ground sites in the legend correspond to Pandora spectrometers installed in the LA Basin and the last five are located in the LMOS domain. Coincidences are identified as the median of all cloud-free GeoTASO TropVCs having pixel centers within a 750 m radius of a Pandora for each overpass (minimum

requirement of 16 valid cloud-free GeoTASO 250 x 250 m pixels for a spatial coverage of at least 1 $km^2$) and the Pandora column observed closest in time to the GeoTASO overpass (within $\pm$ 5 minutes). A Pandora TropVC is derived by subtracting the stratospheric vertical column calculated with the PRATMO climatology from the Pandora total column, following the same approach used with GeoTASO TropVCs. Coincidences without at least two valid Pandora data points within $\pm$ 5 minutes of the GeoTASO overpass are excluded because acquisition of only a single Pandora data point within

ten minutes indicates likely periodic clouds or poor solar tracking by the instrument. TropVCs from GeoTASO compare well with Pandora, with a high correlation ($r^2 = 0.91$), a slope of 1.03, and an offset of $0.52 \times 10^{15}$ molecules $cm^{-2}$.

The whiskers in Fig. 3 indicate the 10th and 90th percentiles of GeoTASO TropVCs within the 750 m radius for GeoTASO (vertical whiskers) and the maximum and minimum Pandora TropVC within $\pm$ 5 minutes from the GeoTASO overpass (horizontal whiskers, representing 6-7 valid Pandora measurements for most of these coincidences), providing a

glimpse of the spatiotemporal variability at the time of the coincidence. Generally, spatial variability (vertical whiskers) increases as the magnitude of $NO_2$ increases. An exception to this pattern occurs at the Zion site. Zion has the lowest surface

albedo of any site, ranging from 1.3-2.6% depending on SZA. The lower albedo over the Zion region together with the relatively low $NO_2$ amounts lead to a lower signal-to-noise ratio in the GeoTASO observations, resulting in a higher uncertainty in the DOAS spectral fit (as discussed in Sect. 2.1). Therefore, the large vertical whiskers at this site are indicative of increased uncertainty in the retrieval in addition to spatial heterogeneity.

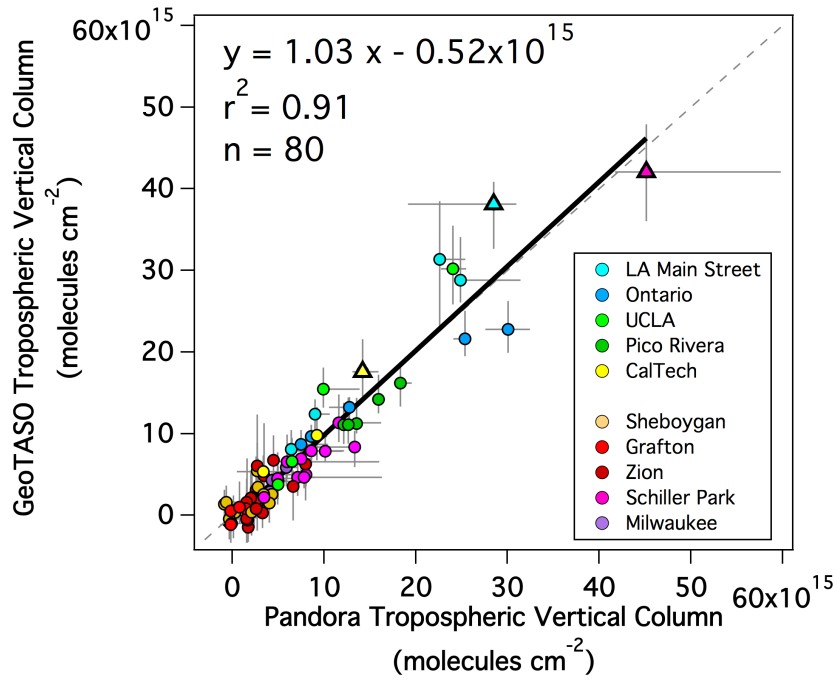

**Figure 3: Scatter plot of GeoTASO $NO_2$ TropVCs vs. Pandora $NO_2$ TropVCs colored by Pandora site. Vertical whiskers show the 90th and 10th percentiles of GeoTASO TropVCs within the 750 m radius of the Pandora site. Horizontal whiskers show the maximum and minimum Pandora TropVCs within the ± 5-minute coincidence window. Triangles indicate coincidences discussed within Sect. 3.1. The grey dashed line indicates the 1:1 line.**

The largest temporal variability (horizontal whiskers) occurs at the Schiller Park site in Chicago, IL, which is located along a major highway near the end of a Chicago O'Hare airport runway. High temporal variability is common at the site and is likely due to local emissions associated with air traffic and nearby roadways. The most polluted point of all coincidences occurs at Schiller Park on June 1st, 2017 (pink triangle in Fig. 3) and this case is further explored in Fig. 4. Figure 4(a) shows a map of the GeoTASO $NO_2$ TropVCs and a hexagon depicting the Pandora location which is colored by the $NO_2$ column observed by Pandora nearest in time to the GeoTASO overpass. The 750 m radius used to define coincident GeoTASO data and an arrow depicting the direction in which Pandora is observing are overlaid. GeoTASO observed a median of $42 \times 10^{15}$ molecules $cm^{-2}$, whereas the Pandora observed $45 \times 10^{15}$ molecules $cm^{-2}$. Inside the 750 m radius, $NO_2$ columns range between 30 and $50 \times 10^{15}$ molecules $cm^{-2}$, and the highest columns observed occur along the viewing direction of Pandora. Figure 4(b) shows a time series of the Pandora TropVC surrounding this overpass and the GeoTASO coincidence. Within an hour prior to the overpass, large temporal variations in $NO_2$ column are observed by the Pandora, suggestive of small-scale plumes influencing the site. In the 10-minute Pandora window (grey section), Pandora peaks at over $60 \times 10^{15}$ molecules $cm^{-2}$. If any temporal averaging is applied to Pandora data in this case, the coincidence would yield a larger apparent difference between GeoTASO and Pandora. The tremendous amount of variability at this highly heterogeneous location suggests caution for use in applications such as satellite validation.

Although the 80 GeoTASO/Pandora coincidences are highly correlated with a slope near 1:1, there are a few outliers that deviate from the linear regression line by amounts larger than the range of observed variability by either instrument. Some of these outliers can be attributed to mismatches in spatial representativeness, as the viewing geometry of the Pandora spectrometer is always oriented in the direction of the sun while the GeoTASO median is constructed from

pixels within the 750 m radius centered on the Pandora location extending into all directions.  Figure 5 highlights one of these cases (yellow triangle in Fig. 3).   Over the CalTech site in the LA Basin on June 27th, 2017, Pandora observes a column of approximately $14 \times 10^{15}$ molecules cm$^{-2}$, whereas the GeoTASO spatial median is $18 \times 10^{15}$ molecules cm$^{-2}$.  The GeoTASO TropVC map (Fig. 5 (a)) shows that Pandora is observing in a direction where NO2 values are at a local minimum, such that the GeoTASO median within the 750 m radius is about 30% higher than Pandora, though GeoTASO TropVCs along the viewing direction of Pandora at that time are more similar to the Pandora observations.  Pandora TropVCs are varying slowly in time surrounding the GeoTASO overpass (Fig. 5(b)). Together these characteristics suggest that the differences between these two observations are more likely associated with the spatial criteria assumed for this comparison rather than the temporal variability near the coincidence time.  Some of the other coincidences showing notable differences, including the two most polluted Ontario points (both lying below the 1:1 line with concentrations of $20\text{-}25 \times 10^{15}$ molecules cm$^{-2}$) also appear to be associated with the assumption in spatial criteria (not shown). Future comparisons of Pandora direct-sun retrievals to high-spatial resolution airborne data could consider the Pandora's viewing geometry in the coincidence criteria, but this is not a viable option for validating satellite products that are not as spatially refined

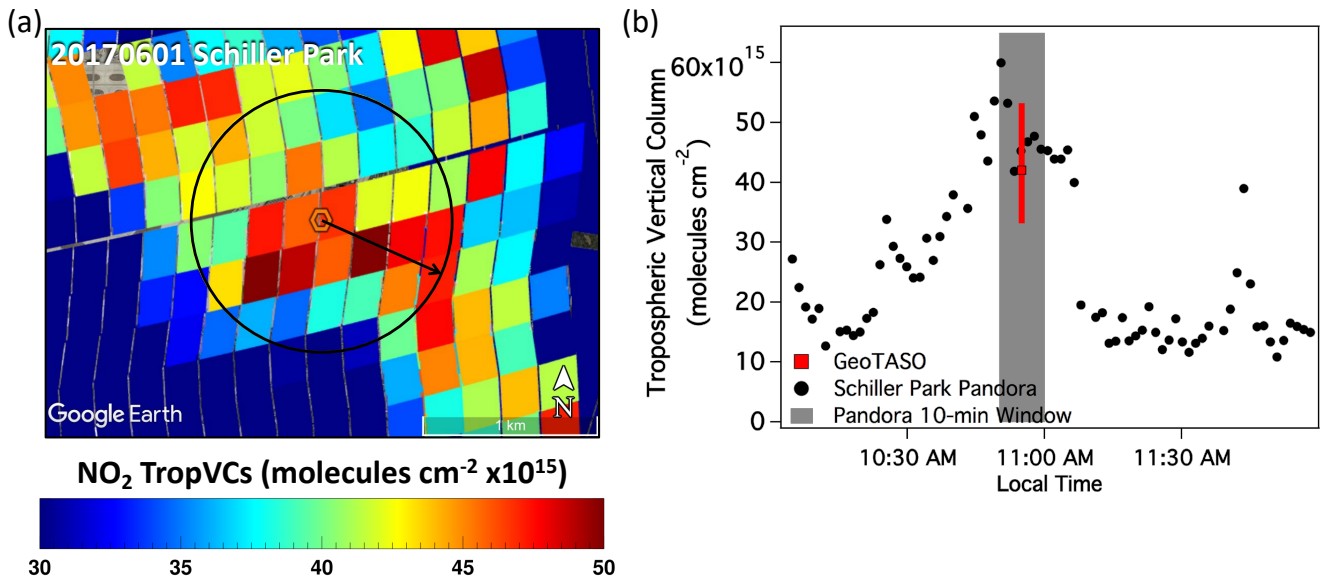

**Figure 4: (a) Map of GeoTASO TropVCs on a linear color scale for the Schiller Park overflight on June 1st, 2017 at 15:55 UTC (10:55 LDT) (pink triangle in Figure 3) with the 750 m radius considered in the spatial binning of GeoTASO overlaid and an arrow depicting the Pandora viewing direction (solar azimuth angle) during the overpass time. The Pandora hexagon is colored by the NO₂ TropVC measured by Pandora during the overpass. (b) Time series showing Pandora data (black points) within approximately ± 1 hour of the GeoTASO overpass.  The Pandora temporal window for the coincidence is shaded in grey and the GeoTASO TropVC and 10th-90th percentiles from the overpass are shown in red.**

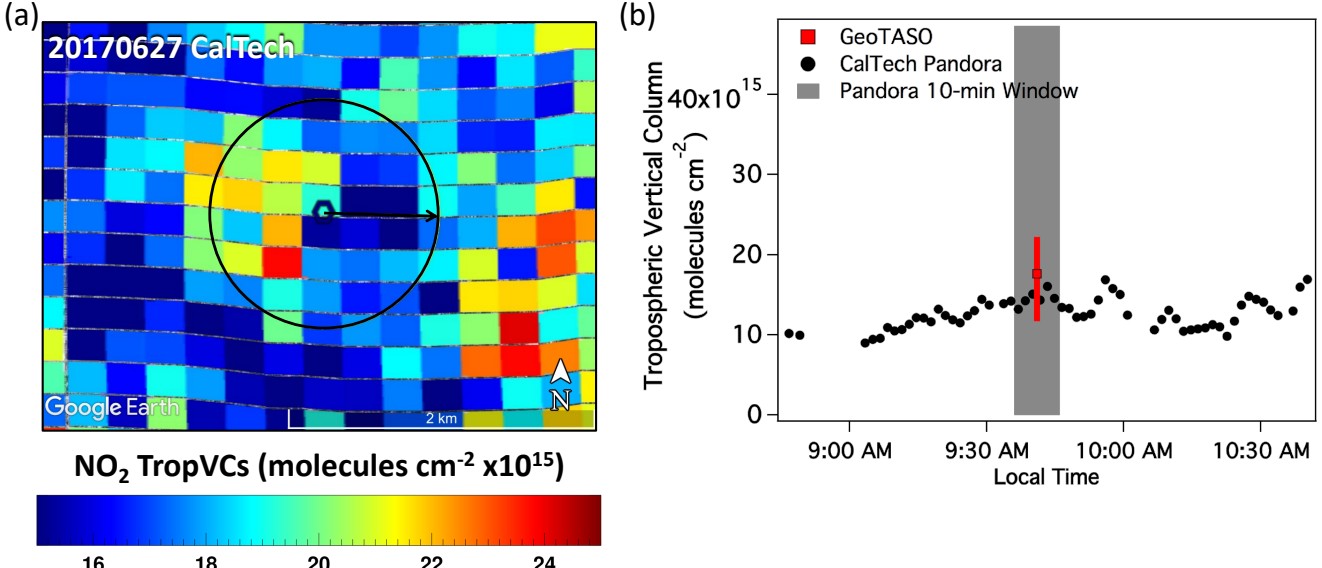

**Figure 5: (a) Map of GeoTASO TropVCs on a linear color scale for the CalTech overflight on June 27th, 2017 at 16:41 UTC (09:41 LDT) (yellow triangle in Figure 3) with the 750 m radius considered in the spatial binning of GeoTASO overlaid and an arrow depicting the Pandora viewing direction (solar azimuth angle) during the overpass time. The Pandora hexagon is colored by the NO₂ TropVC measured by Pandora during the overpass. (b) Time series showing Pandora data (black points) within approximately ± 1 hour of the GeoTASO overpass. The Pandora temporal window for the coincidence is shaded in grey and the GeoTASO TropVC and 10th-90th percentiles from the overpass are shown in red.**

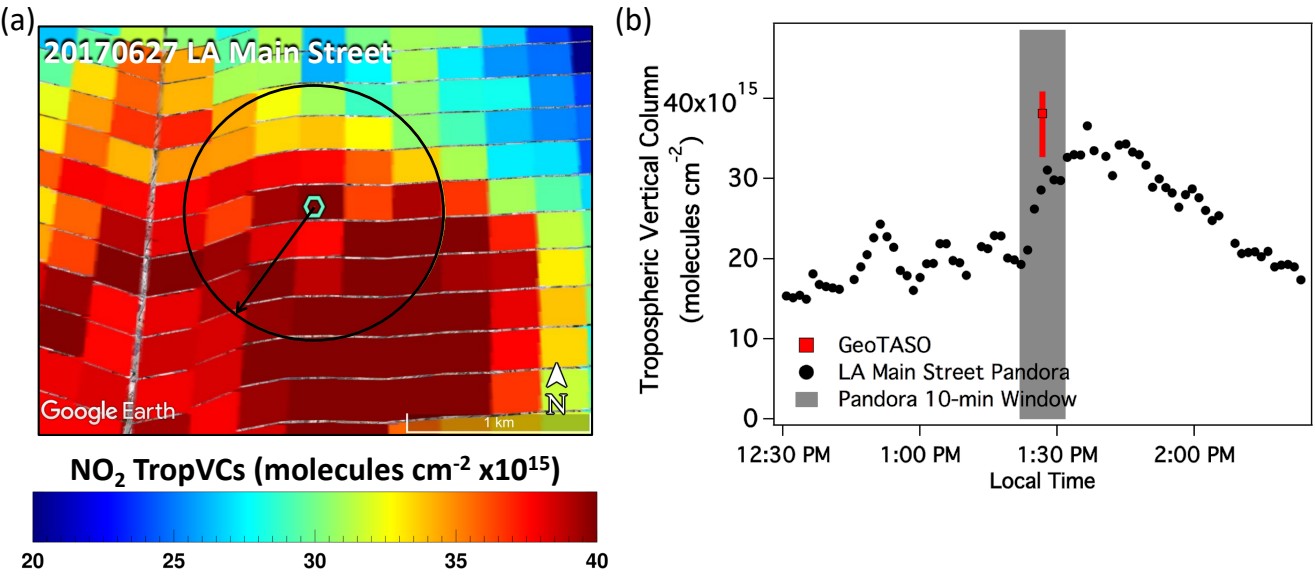

**Figure 6: (a) Map of GeoTASO TropVCs on a linear color scale for the LA Main Street overflight on June 27th, 2017 at 20:26 UTC (13:26 LDT) (cyan triangle in Figure 3) with the 750 m radius considered in the spatial binning of GeoTASO overlaid and an arrow depicting the Pandora viewing direction (solar azimuth angle). The Pandora hexagon is colored by the NO₂ TropVC measured by Pandora during the overpass. (b) Time series showing Pandora data (black points) within approximately ± 1 hour of the GeoTASO overpass. The Pandora temporal window for the coincidence is shaded in grey and the GeoTASO TropVC and 10th-90th percentiles from the overpass are shown in red.**

The largest outlier from the regression line of Pandora-to-GeoTASO coincidences occurs at LA Main Street on the afternoon of June 27th, 2017 (blue triangle in Fig. 3), where GeoTASO observes a TropVC of $38 \times 10^{15}$ molecules cm$^{-2}$ and the Pandora TropVC is only $28 \times 10^{15}$ molecules cm$^{-2}$. This coincidence also shows large gradients both spatially and temporally. What is unique about this case is the environmental/meteorological conditions present during this time period

that make data comparisons and GeoTASO retrieval challenging. Figure 6 has the same formatting as Fig. 4 and Fig. 5. Focusing on the ten-minute window used in the temporal matching of Pandora data, Pandora NO$_2$ TropVC increases from $20 \times 10^{15}$ molecules cm$^{-2}$ to over $30 \times 10^{15}$ molecules cm$^{-2}$. This feature is caused by the arrival of a sea breeze front passing through from the southwest at this time that acts as a convergence zone accumulating NO$_2$ along this front as it passes through the LA Basin (Judd et al., 2018). In this case, because the Pandora data show a rapid increase, any temporal

averaging window placed on the Pandora data for data comparison would not be representative of the conditions during the overpass (similar to the coincidence shown in Fig. 4). Pandora reaches a peak within this 10-minute window of $31 \times 10^{15}$ molecules cm$^{-2}$ approximately one minute after the GeoTASO overpasses, which is closer to the magnitude observed by GeoTASO but still about $7 \times 10^{15}$ molecules cm$^{-2}$ lower. Within 5-10 minutes after the overpass, Pandora values rise again to a range of $33$-$37 \times 10^{15}$, almost as large as GeoTASO median value.

Figure 6(a) indicates that directionality does not reconcile the observations as GeoTASO NO$_2$ TropVCs in the direction Pandora is viewing are consistently larger than Pandora observations, suggesting a potential high bias in the GeoTASO TropVC in this case. High bias could be caused by an underestimation of the AMF; the tropospheric AMF would have to increase from 1.3 to ~1.75 for this area to yield a TropVC similar to Pandora. Given the unique circumstances of the sea breeze front, and the relatively coarse resolution of the NAM-CMAQ model used to obtain the NO$_2$ profile (12 km), we

speculate that perhaps the shape factor used in the AMF calculation may not be representative of the actual NO$_2$ vertical profile within the sea breeze circulation during this time. The rapid fluctuations in Pandora observations through this period also suggest that the airmass is not well mixed, i.e., eddies likely exist in the vicinity of this front which could affect vertical structure at very localized scales. Another contribution could be inaccuracies in the AMF associated with surface reflectance at this site. All LA Main Street coincidences lie above the 1:1 line in Fig. 3 (GeoTASO TropVC is consistently larger than

Pandora), which could be caused by a low bias in the BRDF weighting kernels. LA Main Street is in an area that has fewer valid retrievals from the MCD43A1 product during the monthly average used for BRDF input to VLIDORT. Further investigation of systematic biases by site due to BRDF uncertainty would require more consistent sampling at the site, which will be possible with satellite products and long-term Pandora measurements used for product validation in the future.

**Table 3: Statistics between Pandora and the GeoTASO data at the nominal pixel size shown in Figure 3 as well as the simulated pixel**
**sizes for TEMPO, TROPOMI, and OMI products shown in Figure 9.**

| Pixel Size | All Points | | | | Pandora < $40 \times 10^{15}$ molecules cm$^{-2}$ | | | | Pandora < $20 \times 10^{15}$ molecules cm$^{-2}$ | | | |
|---|---|---|---|---|---|---|---|---|---|---|---|---|
| | slope | Intercept x10$^{15}$ | r$^2$ | N | slope | Intercept x10$^{15}$ | r$^2$ | N | slope | Intercept x10$^{15}$ | r$^2$ | N |
| Nominal: | 1.03 | 0.52 | 0.91 | 80 | 1.08 | -0.75 | 0.89 | 79 | 0.96 | -0.30 | 0.84 | 73 |
| TEMPO: 3 x 3 km | 0.88 | -0.07 | 0.86 | 79 | 1.03 | -0.72 | 0.89 | 78 | 0.93 | -0.35 | 0.85 | 72 |
| TROPOMI: 5 x 5 km | 0.77 | 0.11 | 0.88 | 76 | 0.88 | -0.32 | 0.89 | 75 | 0.87 | -0.36 | 0.81 | 69 |
| OMI: 18 x 18 km | 0.57 | 0.9 | 0.61 | 66 | 0.73 | 0.01 | 0.83 | 65 | 0.89 | -0.7 | 0.78 | 60 |

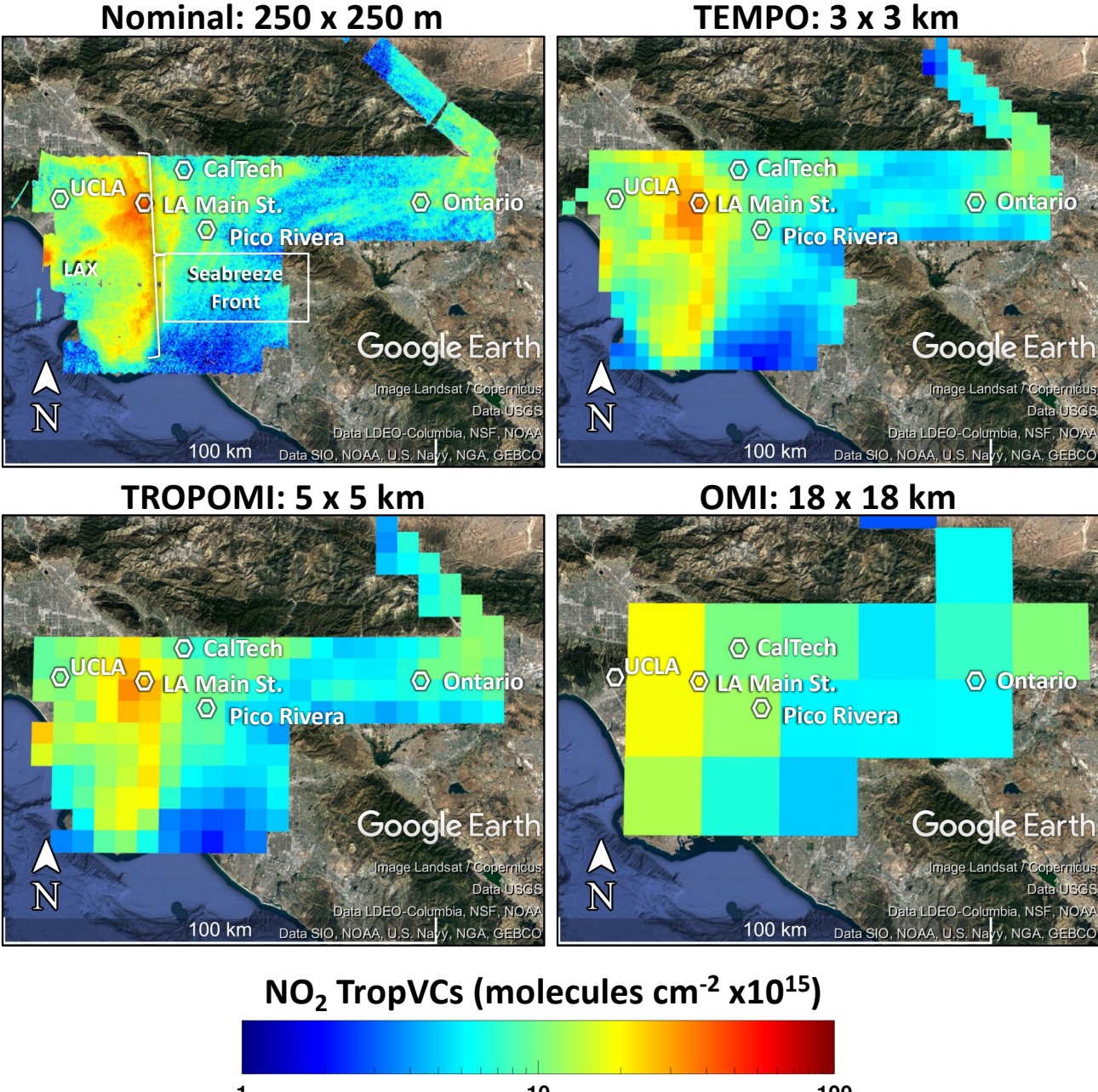

**Figure 7: Maps of GeoTASO NO₂ TropVC on a log10 color scale averaged to the labeled pixel sizes to demonstrate the spatial detail that would be observed from the midday raster in Los Angeles on June 27ᵗʰ, 2017. All pixels shown have at least 30% of their area mapped by GeoTASO during this timeframe.**

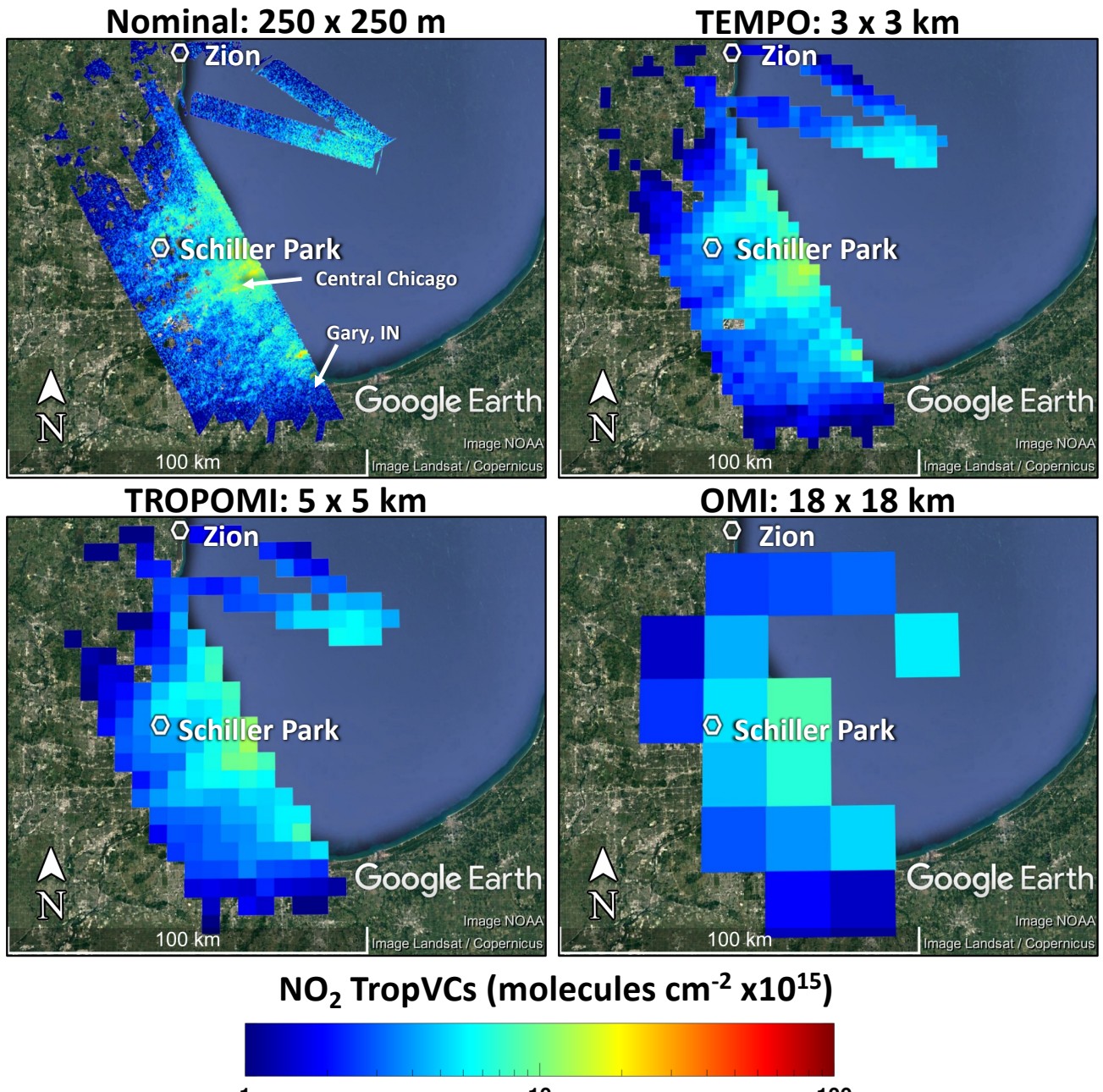

**Figure 8: Maps of GeoTASO NO₂ TropVC on a log10 color scale averaged to the labeled pixel sizes to demonstrate the spatial detail that would be observed from the midday in the Chicago area on May 22nd, 2017. All pixels shown have at least 30% of their area mapped by GeoTASO during this timeframe.**

## 3.2 Scaling GeoTASO to satellite product footprints

Past, present, and planned satellite instruments have spatial resolutions that are coarser than the GeoTASO airborne measurements. The gapless mapping strategy executed with GeoTASO during summer 2017 provides fine-resolution data suitable for averaging to coarser areal resolutions typical of these current and next-generation satellite retrievals. As demonstrated by the good agreement between Pandora and GeoTASO shown in Sect 3.1, the GeoTASO observations during these rasters capture the spatiotemporal variability existing in these areas. Comparisons of upscaled GeoTASO retrievals with Pandora measurements provide an early assessment of the capabilities of the next-generation sensors for resolving urban scale $NO_2$ and also give insight toward validation strategies in urban regions, where sub-pixel variability within the satellite product footprint may be significant.

To simulate the satellite products, each GeoTASO raster is averaged to fixed grids with pixel areas representative of near-nadir observations from TEMPO, TROPOMI, and OMI. These grids are simplified by making each pixel square in shape to avoid introducing an orientation bias. Simulated pixels are 3 x 3 km (9 km$^2$) for TEMPO (literature reported 9.24 km$^2$ per pixel, Zoogman et al., 2017), 5 x 5 km (25 km$^2$) for TROPOMI (literature reported 24.5 km$^2$, van Geffen et al., 2019), and 18 x 18 km (324 km$^2$) for OMI (FoV75Area for nadir is 338.4 km$^2$).

GeoTASO data are remapped to these fixed grids by computing a weighted average of the GeoTASO TropVCs based on the areal overlap within each simulated satellite grid pixel for each raster. Figure 7 shows an example of this process for the midday Los Angeles raster on June 27th, 2017, at the nominal GeoTASO resolution and each simulated satellite product resolution. At this time, a marine/land airmass convergence zone was created by the development of a sea breeze that resulted in an accumulation of $NO_2$ seen as a line of enhanced $NO_2$ in excess of 30x10$^{15}$ molecules cm$^{-2}$ extending north and south of LA Main Street (discussion of Fig. 6, Sec. 3.1; Judd et al., 2018). A plume is also seen extending inland from the Los Angeles International Airport (LAX). As expected, fine scale features evident in the native GeoTASO observations are increasingly blurred as the spatial resolution is degraded. The airport plume is spatially distinct up to the TEMPO spatial scale and the sea breeze is resolved through the TROPOMI spatial scale, but neither are visible at the spatial scale of OMI.

Another example is shown in Fig. 8 for a Chicago raster near midday on May 22nd, 2017. In this example, the southwesterly wind throughout the day quickly advected local emissions out over Lake Michigan, allowing visualization of fresh emission source regions and broadening and accumulation of the resulting pollution plume as it passes over the Chicago metropolitan area. The native-resolution GeoTASO retrievals show three areas of enhanced $NO_2$: near Schiller Park-O'Hare Airport, central Chicago, and the industrial shore near South Chicago and Gary, IN. This raster was repeated 3 other times on this day and all four rasters show a similar spatial pattern but magnitude of $NO_2$ TropVCs decreases throughout the day, likely indicating that morning emissions are larger than afternoon emissions on this day (not shown). These spatial features are discernible up through the areal resolution of TROPOMI but the OMI spatial scale is larger than the physical separation of these three enhanced regions leading to loss of this spatial information. The maximum TropVC observed at an OMI spatial resolution is even displaced, such that it does not accurately represent the location of the highest $NO_2$ TropVCs.

This upscaling technique was applied to all GeoTASO flights listed in Table 1. To assess the impact of spatial resolution on comparisons to Pandora, each Pandora coincidence from Sect. 3.1 is matched to its equivalent upscaled pixel. Fig. 9 shows the scatter plots of these coincidences and the associated linear regression statistics are shown in Table 3. To avoid excess extrapolation, only upscaled grid cells that are at least 30% sampled by GeoTASO are considered in this analysis, which is why the number of coincidences decrease as pixel size increases (Table 3).

Increasing pixel size progressively worsens 1:1 Pandora agreement with the slope decreasing from 1.03 at native resolution to 0.88, 0.77, and 0.57, respectively at TEMPO, TROPOMI, and OMI scales. The decrease in slope is partially driven by the most polluted coincidence at Schiller Park. Figure 10 shows this coincidence in context (zoomed out map from Fig. 4) with the upscaled equivalent pixels from TEMPO, TROPOMI, and OMI overlaid. The two nearest TEMPO pixels are shown because the Pandora location is near a pixel edge, just inside the left TEMPO pixel. The GeoTASO observations show that the Pandora is located within a narrow region of strongly enhanced $NO_2$. The size of the feature observed by GeoTASO and Pandora in this case is smaller than the spatial scale of TEMPO pixels, and because the TEMPO pixels also include regions of much lower $NO_2$ values, the simulated-satellite values decrease from 42x10$^{15}$ molecules cm$^{-2}$ at native GeoTASO resolution to less than 30x10$^{15}$ molecules cm$^{-2}$ at simulated TEMPO resolution. This case illustrates the challenge

of determining the difference between retrieval bias and spatial misrepresentation but also the advantage of having high-spatial resolution airborne spectrometer observations as a transfer standard between satellite measurements and localized column measurements from Pandora. In this analysis, the native resolution retrieval does not appear to be biased (Sect. 3.1). Therefore, the changes in the relationship between upscaled GeoTASO observations and Pandora are solely due to spatial

5    representation.  In the case of this coincidence with its high degree of spatial and temporal variability, none of the simulated satellite products are capable of resolving this dominate sub-grid scale feature observed by GeoTASO and Pandora, and without careful analysis the satellite products could appear to have a low bias if extrapolating validation datasets are extended to include column amounts that are likely representative of extreme sub-grid features too large for the satellite retrieval to resolve.

10        Figure 9 illustrates that a linear relationship appears to degrade above some threshold value of $NO_2$ in these cases. Above the threshold, the more polluted $NO_2$ columns observed by Pandora are likely occurring over spatial scales smaller than the satellite resolutions, as shown in the preceding example. This theoretical threshold decreases as the spatial resolution is coarsened.  The decreasing linear slope with increasing pixel size simply results from this flattening of the satellite-to-Pandora relationship at the higher $NO_2$ values. This threshold does not imply that the satellite cannot detect such

15    large amounts of pollution, but instead reflects that in these regions $NO_2$ enhancements of these large magnitudes are not sustained over spatial scales that are representative of current/future space-based retrievals. Such behavior makes sense if the high $NO_2$ values are associated with localized heterogeneous features rather than more mixed regional-scale enhancements, which is a reasonable conceptual model for urban areas such as these. The value of the threshold likely varies by region and may not be generalizable, but these results suggest that there is a limit to the magnitude of pollution that can

20    be validated using the localized measurements of Pandora (i.e. Pandora is only representative of each satellite product up to a certain pollution scale). As longer-term Pandora data records become available, it is possible that examining temporal variance statistics at each site can allow development of screening criteria to help identify such situations.

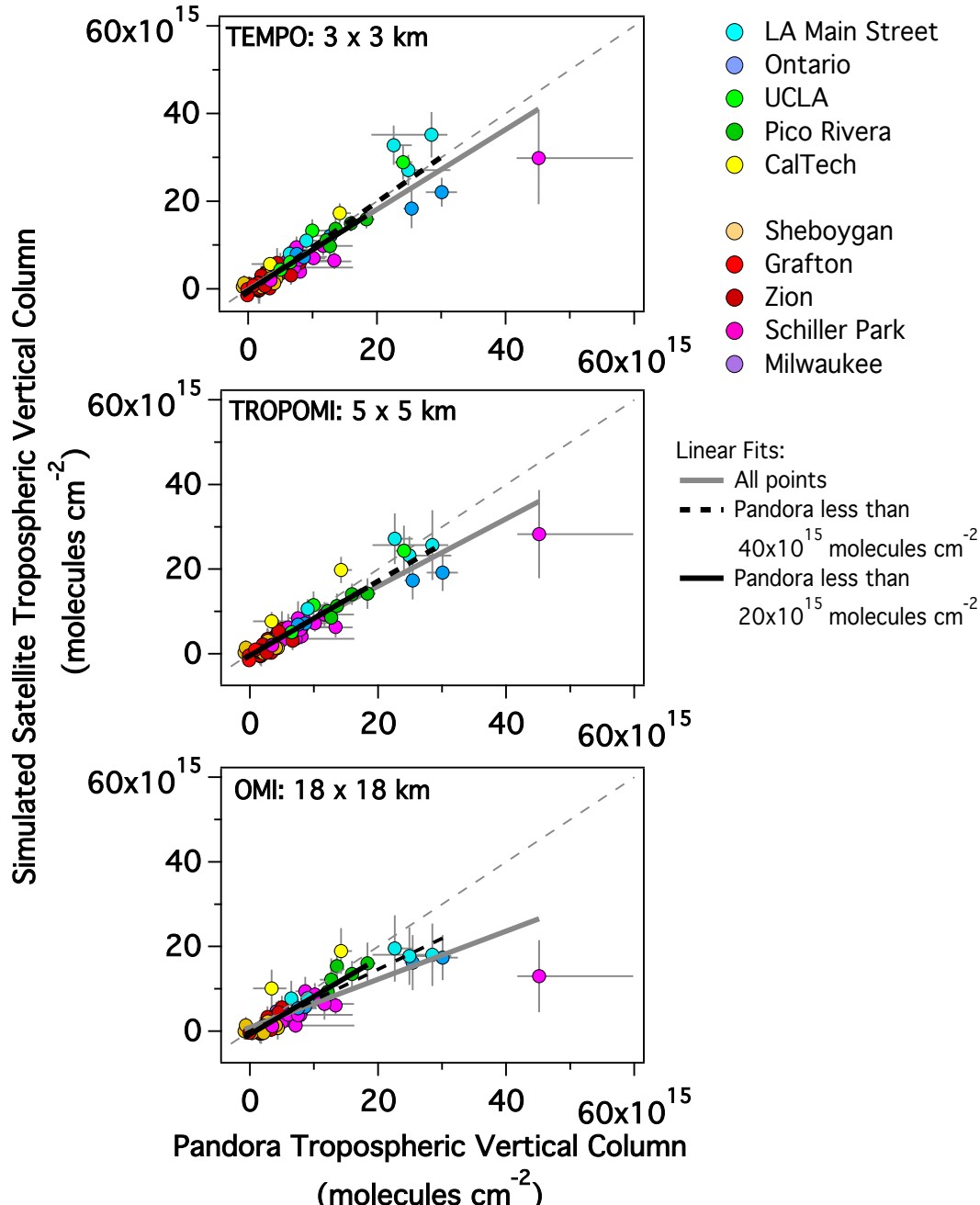

**Figure 9: Scatter plot of GeoTASO TropVCs scaled to the nadir areal resolution of TEMPO (a), TROPOMI (b), and OMI (c) TropVCs vs. Pandora TropVCs colored by site and their associated linear fits (statistics listed in Table 3). Vertical bars show standard deviation of GeoTASO TropVCs within the upscaled pixel. Horizontal bars show the minimum and maximum of Pandora data within the ± 5-minute coincidence window. The grey dashed line indicates the 1:1 line.**

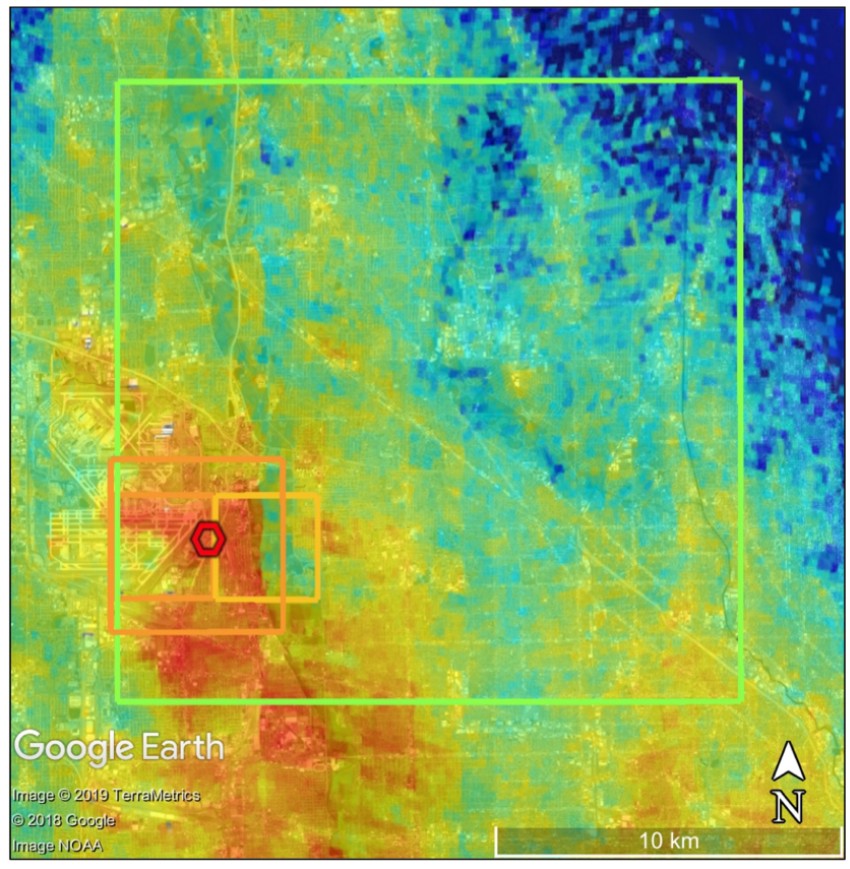

**NO₂ TropVCs x10¹⁵ molecules cm⁻²**

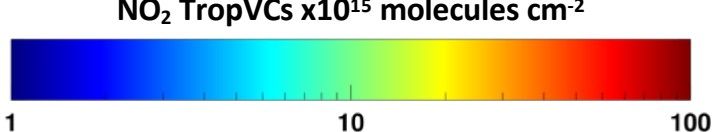

**Figure 10: Map of GeoTASO TropVCs on a log10 color scale at the 250 x 250 m spatial resolution during the morning raster of June 1st, 2017 over Schiller Park (same as Figure 4) with the overlaid simulated pixels for TEMPO, TROPOMI, and OMI encompassing the Schiller Park ground site colored by the magnitude of their upscaled TropVC. The hexagon indicates the location of the Pandora and is colored by the observed NO₂ from the Pandora.**

The data from these campaigns are not sufficient for confidently quantifying such thresholds but do allow the sensitivity to different thresholds to be explored by examining how the statistics vary as pollution thresholds for Pandora coincidences are decreased. To do this, each regression is repeated by excluding points for which Pandora TropVCs exceed $40\times10^{15}$ molecules cm⁻² (dashed regression line in Fig. 9) and exceed $20\times10^{15}$ molecules cm⁻² (solid black regression line in Fig. 9). Only the most polluted Schiller Park coincidence is excluded when filtering by Pandora TropVCs less than $40\times10^{15}$ molecules cm⁻². The example discussed in association with Fig. 10 shows that this is a case in which the Pandora and GeoTASO are observing a highly localized feature. With the removal of this point, the slopes increase (improve) and the correlations remain similar, with the exception of a notable improvement at OMI resolution. The TEMPO scale comparison resembles the results at the nominal spatial scale, but slope still decreases as data are scaled to the coarser TROPOMI and OMI resolutions. When considering coincidences limited to Pandora TropVCs less than $20\times10^{15}$ molecules cm⁻², the

correlations all degrade by 5-10% due to the smaller dynamic range with similar noise but are still on order of 0.8 or better. Pixel size has less influence on calculated slope, with only a 7% decrease in slope between the nominal pixel size and the OMI scale resolutions. Even below these thresholds, individual coincidences are often associated with influence of spatial heterogeneity as shown by the vertical whiskers.  In fact, no single Pandora site was completely homogeneous for all coincidences (determined by the standard deviation of the GeoTASO measurements in the area surrounding each site as well as visual examination of Pandora time-series near the times of coincidences).  However, if this type of pollution environment is sampled enough, the variability could be statistically characterized sufficiently enough to allow identification of retrieval biases or errors.

This analysis shows that, if a priori information at high enough spatial resolution is applied in the satellite retrieval, Pandora and satellite scale products can compare well statistically within these regions up to the nadir pixel size of OMI for Pandora columns less than $20 \times 10^{15}$ molecules cm$^{-2}$ (although individual points can still deviate largely from the 1:1 relation due to sub-pixel variability).  The threshold pollution scale for TROPOMI comparisons to Pandora is higher, somewhere between $20\text{-}40 \times 10^{15}$ molecules cm$^{-2}$, and for TEMPO is higher still, up to $40 \times 10^{15}$ molecules cm$^{-2}$.  This analysis would benefit from more coincidences in the range of $20\text{-}60 \times 10^{15}$ molecules cm$^{-2}$ to more confidently define pollution scale thresholds acceptable for Pandora and satellite product comparisons.  Acceptable pollution ranges for data comparison may even vary from region-to-region as each urban area could have distinct spatial variability patterns.

These results suggest that Pandora spectrometer direct-sun observations are very useful for validating satellite NO$_2$ products at the spatial resolutions of TEMPO and TROPOMI, as the products are highly correlated with points tightly spanning both sides of the 1:1 line for values of up to approximately $30 \times 10^{15}$ molecules cm$^{-2}$ (and perhaps higher, but this dataset lacks sufficient observations above that level).  However, even at the finest satellite product resolution (TEMPO), comparisons with Pandora observations are not exempt from outliers caused by extreme degrees of subpixel heterogeneity. When choosing validation locations, areas known to have consistently strong spatiotemporal variability at spatial scales finer than the satellite retrieval (e.g., Schiller Park) should be avoided. When possible, additional measurements (i.e., high resolution airborne mapping) would help investigate uncertainty associated with spatial representativeness in the analysis of retrieval bias.

### 3.3 Comparison of OMI satellite products and ground-based Pandora measurements

During the LMOS and LA Basin flights, OMI was the highest resolution space-based platform in orbit, as TROPOMI was not launched until October of that year. Since 2004, OMI has been observing global NO$_2$ columns at a regional spatial scale on daily timescales.  The simulated OMI comparisons in Sect. 3.2 are considered an idealized 'best case' scenario for data comparisons due to the near-nadir spatial resolution and the high resolution a priori used for GeoTASO AMFs. In actuality, OMI's pixel area increases over 1200% from nadir to the swath edge and most of the a priori inputs to the AMF calculations have coarser spatial resolutions than those used in these GeoTASO retrievals. This section demonstrates how actual OMI observations compare with the Pandora measurements in the LMOS and LA Basin domains during May-July 2017 to cover the time period during which OMI BEHR data were available and the Pandoras were operating in the two regions.

Figure 11 shows the 444 coincidences between the OMI BEHR TropVC and the Pandora TropVC (Pandora total column minus the OMI stratospheric column) with the best-fit linear regression for all points shown as a dashed black line. The points are further subset by considering only pixel areas less than 1000 km$^2$ (black points and solid black regression line).  Coincidence and comparison criteria include OMI pixels unaffected by the row anomaly encompassing the location of a Pandora with a MODIS cloud fraction less than 20%. Pandora data are averaged within a $\pm$ 5 min window of the OMI overpass time to be consistent with the temporal window used to assess variability in Sect 3.1.

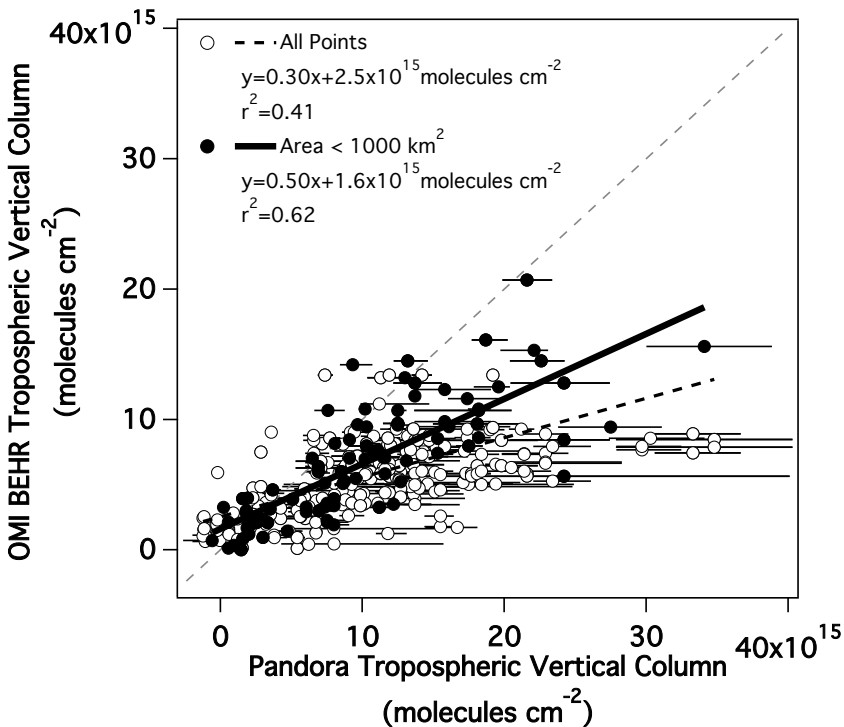

**Figure 11: OMI Native Pixel BEHR NO₂ tropospheric vertical column data vs. Pandora NO₂ tropospheric column. The filled circles indicate pixel areas less than 1000 km². The linear fit to all points is shown in dashed black and the linear fit for pixels less than 1000 km² is shown in solid black. Horizontal whiskers show the maximum and minimum of Pandora data ± 5 minutes of the OMI overpass. The thin dashed line is the 1:1 line.**

Considering all pixel areas, the BEHR TropVCs are moderately correlated with Pandora TropVCs ($r^2 = 0.41$) with a low slope (0.30) and are on average biased 50% lower than Pandora. Considering only pixels with areas less than 1000 km², correlation and slope improve by at least 50% ($r^2 = 0.62$, slope=0.5), again demonstrating the sensitivity of these direct comparisons to areal representativeness of the satellite pixel. Restricting OMI pixels to even smaller sizes (not shown) does not result in further statistical improvement, probably because the number of coincidences quickly decreases as the areal threshold decreases (40 coincidences when including only pixel areas below 500 km² and 23 coincidences for below 400 km²). The slopes shown in Figure 11 are notably lower than those calculated from the idealized GeoTASO-simulated idealized nadir OMI pixels discussed in Sect. 3.2, by 58% for all size pixels and 30% for pixel area < 1000 km². These differences are indicative of improvements associated with using higher spatial resolution a priori information, as used in the GeoTASO retrievals.

The horizontal whiskers in Figure 11 show the maximum and minimum in the Pandora measurement within the 10-minute temporal window of the OMI overpass. Some of these coincidences are impacted by very large variations in NO₂ measured by Pandora within a small window of the OMI overpass. This suggests that in addition to the magnitude of NO₂ as a potential heterogeneity screen discussed in Sect. 3.2, coincidence criteria based on the temporal variability of the fast measurements from Pandora could also help identify coincidences that may be impacted by small-scale highly heterogeneous environments.

To further explore sensitivity to AMF, Table 4 lists the linear regression statistics of OMI retrievals against Pandora observations for three products: the BEHR vertical column data shown in Fig. 11, the NASA Standard Product V3 vertical column (SP TropVC), and tropospheric slant column products from OMI and Pandora (TropSC) (Recall that BEHR also uses the SP slant column). Consistently comparing these slant and vertical column statistics with respect to Pandora observations provides an assessment of the influence of AMF a priori assumptions. For these Pandora coincidences, SP

TropVCs are on average 51% lower than BEHR TropVCs, resulting in a 40% lower slope for SP TropVCs and Pandora. These comparisons demonstrate improvement due to incorporating the higher resolution a priori inputs in the BEHR product. Improvements to the spatial resolution of a priori input for OMI AMF calculations have previously been shown to help reduce biases (Russell et al., 2011; Lin et al., 2014). There is a minimal difference in $r^2$ when comparing Pandora to BEHR
or SP TropVCs for OMI pixels less than 1000 km$^2$. But there is a dramatic loss in correlation between SP TropVCs and Pandora as pixel size increases, demonstrating that the impact of coarse a priori input worsens with increasing pixel size.

OMI tropospheric slant column (TropSC) comparisons with Pandora are slightly better than SP vertical column (SP TropVC) comparisons, indicating that the SP AMF is introducing additional variance in the SP TropVC product. The loss in skill associated with applying the SP AMF is evidenced by the 30% decrease in the slope for all coincidences and 22%
decrease for pixels less than 1000 km$^2$ in comparison to TropSC comparisons. The application of the finer-resolution BEHR AMF largely retains the amount of skill in the TropSC with a slight improvement in slope.

Previous comparisons of OMI with Pandora direct-sun column measurements in urban environments have also shown low to moderate correlations for vertical-to-vertical column comparisons with low biases found in areas subject to spatial heterogeneity of $NO_2$ (Lamsal et al., 2014; Reed et al., 2015; Kim et al., 2016; Goldberg et al., 2017). Similar results
have also been obtained from comparisons with ground-based MAX-DOAS retrievals (Lin et al., 2014; Chan et al., 2015). Monthly averaged OMI NASA SP V3 observations were also found to be biased low using Pandora observations in Korea (Herman et al., 2018) and MAX-DOAS observations in Hong Kong (Krotkov et al., 2017), but both studies found consistent seasonal trends in the satellite and surface-based column measurements. Not all comparison studies result in such a strong low bias associated with OMI. Boersma et al. (2018) found no low bias in OMI observations, but care was taken to limit
influence by spatial variability and there were only 31 coincidences ranging from 5-10x10$^{15}$ molecules cm$^{-2}$, which are pollution levels that are likely less influenced by extreme spatial heterogeneity. Lamsal et al. (2014) noted that MAX-DOAS/OMI bias (-16.3%) was not quite as low as reported by the Pandora comparisons in a study in the polluted (up to 40x10$^{15}$ molecules cm$^{-2}$) Tsukubu region of Japan, which may be related to either the larger spatial footprint of MAX-DOAS observations (~10 km from Boersma et al., 2018) or different spatiotemporal thresholds appropriate for that specific region.
For validation purposes, there should be value in combining direct-sun and all-sky (MAX-DOAS) measurements at the same locations. The MAX-DOAS measurements, with their ability to measure omni-directionally, can be more spatially representative albeit at the expense of coarser temporal resolution and higher measurement uncertainties, while the direct-sun observations are more frequent with very low measurement uncertainties but potentially larger influences of local scale variability.
Theoretically, satellite retrieval uncertainty could be better quantified with measurements from airborne simulators like GeoTASO that are capable of mapping entire satellite pixels. During LMOS/LA Basin flights, GeoTASO sampled inside 177 OMI pixels within ± 1 hour of the OMI overpass. However, during these 2017 flights, no OMI pixels were entirely mapped within a ± 1-hour period from the OMI overpass; the maximum OMI pixel area covered was approximately 80% in one coincidence and only 7 pixels were mapped by at least 50%. Therefore, GeoTASO data from these campaigns do
not provide enough data to independently assess OMI bias. However, with the finer spatial resolutions of new platforms (e.g. TROPOMI and TEMPO), mapping satellite pixels within an acceptable temporal threshold from the overpass time should be more easily achieved, allowing an additional point-of-view between the space-based observations and ground-based platforms such as Pandora.



**Table 4: Statistics between Pandora and OMI BEHR and SP TropVC and TropSC based on OMI pixel area.**

| OMI Pixel Area | Slope | | | Intercept (x$10^{15}$ molecules cm$^{-2}$) | | | $r^2$ | | | N |
|---|---|---|---|---|---|---|---|---|---|---|
| | BEHR TropVC | SP TropVC | TropSC | BEHR TropVC | SP TropVC | TropSC | BEHR TropVC | SP TropVC | TropSC | |
| **< 1000 km$^2$** | 0.50 | 0.34 | 0.44 | 1.6 | 1.2 | 1.3 | 0.62 | 0.61 | 0.65 | 125 |
| **All** | 0.30 | 0.18 | 0.26 | 2.5 | 1.9 | 2.4 | 0.41 | 0.32 | 0.39 | 444 |

## 4 Conclusions

During May-June 2017, an observing strategy was executed to build a number of airborne high spatial resolution gapless maps with GeoTASO over networks of ground-based instruments along the western shore of Lake Michigan and in the LA Basin. Each region had a network of five Pandora spectrometers (10 total) operating in direct-sun mode providing accurate NO$_2$ vertical column measurements at high temporal resolution with which to compare GeoTASO NO$_2$ TropVC retrievals. The sub-kilometer airborne NO$_2$ retrievals from GeoTASO are highly correlated with Pandora spectrometer observations with a slope near 1:1. Most of the apparent discrepancies between GeoTASO and Pandora TropVCs are associated with high variability and are therefore sensitive to assumptions made for identifying coincidences: e.g., the uni-directional Pandora viewing footprint vs. the omni-directional 750 m radius used to subset GeoTASO. At least one incidence was from a likely GeoTASO TropVC bias induced by inaccurate a priori in a complex environment that is difficult to simulate (a sea breeze frontal convergence zone). Future considerations for such comparisons should include assessments based on temporal and spatial criteria, as well as different environmental conditions. Despite these individual discrepancies, the GeoTASO retrievals accurately capture the spatiotemporal variability of NO$_2$ in these regions.

The raster mapping strategy at sub-kilometer resolution allows assessment of the impacts of sub-pixel heterogeneity within the larger areal pixel resolutions of past, present, and planned satellite instruments. Data from each raster are binned to fixed grids to simulate the pixel areas of nadir observations from TEMPO (3 x 3 km), TROPOMI (5 x 5 km) and OMI (18 x 18 km). Distinct spatial features (sea breeze front, industrial areas, etc.) within urban areas of Los Angeles and Chicago can be distinguished at TEMPO, and perhaps even TROPOMI, scale but cannot be resolved at the nadir area scale of OMI. The coarseness of the OMI scale can even alias the apparent spatial location of NO$_2$ maxima within urban regions. As pixel size increases, the linear statistics with Pandora degrade in these urban regions due to non-resolvable spatial heterogeneity that is most often associated with highly-polluted features that lie within only a portion of the larger pixels. This suggests that for validation purposes, there may be a practical limit on the magnitude of Pandora measurements that can be compared to satellite retrievals as occurrences of exceptionally high concentration NO$_2$ features appear to occur at spatial scales smaller than the satellite resolutions. In the regions of this study, this resolvable pollution scale appears to be near 20x$10^{15}$ molecules cm$^{-2}$ at OMI spatial resolution. TROPOMI and TEMPO resolution satellite products have the capability to compare well with Pandora at higher pollution scales, although more coincidences above 30x$10^{15}$ molecules cm$^{-2}$ would help refine this conclusion further. These thresholds provide only general guidance, as individual coincidences can still be influenced by spatial heterogeneity at any magnitude of pollution. However, if such environments are sampled enough, the variability could be statistically characterized sufficiently enough to allow identification of retrieval biases or errors.

Actual OMI observations show poorer agreement with Pandora direct-sun observations than is indicated by the idealized simulated satellite product. This is partially attributable to the decrease in OMI spatial resolution off-nadir, with some of the coincidences having pixel areas up to 1200% larger than the nadir pixel areas. When BEHR TropVC pixels are filtered to include only areas less than 1000 km$^2$, the Pandora linear regression slope improves by 66% and $r^2$ by 51%. The degraded real-world performance is also due to coarse information in the AMF calculation. An indication of this sensitivity is shown by the vertical column regression statistics against Pandora coincidences, in which the BEHR product has a better

slope than the NASA Standard Product. Further, the regression of slant column products is better than the SP vertical column, showing that uncertainties introduced by coarse AMF assumptions can even degrade the satellite $NO_2$ products.

These results reiterate the past challenges in using OMI observations to evaluate the magnitude of $NO_2$ pollution within urban areas, as the large footprint spatially averages $NO_2$ over an area that in reality has more fine-scale features.

With the launch of TROPOMI in October 2017 and the geostationary sensors arriving in the next decade, their unprecedented spatial and temporal resolutions offer significantly improved capability for accurately assessing pollution within urban areas. Future validation work of satellite retrievals should consider using networks collecting frequent direct-sun Pandora observations spread throughout a wide-range of pollution environments that, through time, can collect enough data to statistically characterize effects of small-scale heterogeneity in order to evaluate satellite retrievals. Assessing

temporal behavior of the Pandora observations (and the Geostationary observations when available) surrounding coincidence times can add additional information about whether or not any resultant difference between the measurements can be associated with spatiotemporal heterogeneity (e.g., the sea breeze front at LA Main Street in Fig. 6 and Fig. 7). Future small-scale campaigns similar to those reported here, combining high spatial resolution airborne mapping observations and high temporal resolution ground-based reference measurements with high-resolution modeling studies, provide a framework

for validation of geostationary air quality measurements that will greatly enhance their usage for science and applications.

## Author Contributions

LMJ processed the GeoTASO $NO_2$ retrievals and led writing of the manuscript. All co-authors provided feedback on the methodology and contributed to the final manuscript. JAA, RBP, and LMJ led flight planning activities for LMOS and/or the LA Basin Flights. SJJ, MGK, and LMJ collected the GeoTASO dataset during LMOS and/or the LA Basin flights, and SJJ

and MGK calibrated and processed the L1b spectra for GeoTASO. RBP provided the NAM-CMAQ analysis used in the vertical column retrieval. JJS, LCV, DW set up and monitored the Pandora spectrometers during LMOS. RS and NA provided the Pandora spectrometers for the LA Basin and NA set up the LA Basin Pandora instruments. AC, MM, and MT processed the Pandora $NO_2$ retrievals.

## Data Availability

All datasets are publicly available online: GeoTASO data are archived at https://www-air.larc.nasa.gov/cgi-bin/ArcView/lmos, Pandora data are found at data.pandonia.net, and OMI data are from Laughner et al. (2018b).

## Acknowledgements

We would like to acknowledge NASA Earth Science Division's GEO-CAPE Mission Study for funding GeoTASO flights, the LMOS Science Teams, Caroline Nowlan and Gonzalo Gonzalez Abad at Harvard SAO for providing knowledge and

support in the use of the SAO AMF calculation tool, the South Coast Air Quality and Monitoring District (SCAQMD) and colleagues at UCLA and CalTech for providing accommodations for the Pandora instruments in the LA Basin, our EPA colleagues and site hosts for maintaining Pandora instruments in the LMOS domain, NASA SARP 2017 and NSRC, Barry Lefer and the NASA Tropospheric Composition Program for inviting us to fly as part of the Student Airborne Research Program in the LA Basin, and the NASA Langley Research Center pilots and flight crew during both field missions.

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
