# Peer review of "Evaluating the impact of spatial resolution on tropospheric NO2 column comparisons within urban areas using high-resolution airborne data"

_Atmospheric Measurement Techniques, 2019_

## Referee Comment (RC1)

The paper studies the impact of typical (relatively coarse) spatial resolutions of past, present and planned satellite missions on tropospheric $NO_2$ retrievals over areas characterized by a strong spatiotemporal variability in the $NO_2$ field. High resolution airborne GEOTASO $NO_2$ VCDs as well as TEMPO, TROPOMI and OMI VCDs, simulated from the GEOTASO observations, are compared with coincident observations at 10 PANDORA sites at the western shore of Lake Michigan and over the Los Angeles Basin. By combination of the different data sets, the work provides an interesting insight into the spatiotemporal tropospheric $NO_2$ variability over polluted areas. The impact of mismatched spatial representation has been quantified. The results and discussions are valuable in the assessment of validation strategies for the future generation of air pollution satellites. The scientific content of the paper fits well within the scope of AMT, and the manuscript is well-written and generally well-structured. However, some revisions (detailed below) should need to be conducted in the paper before publication.

**General comments**

The first two sections are missing some essential information that are valuable to the reader to better interpret the results and the geophysical parameters. I suggest the following:

-Please add a "Campaign" section. Some information about the campaigns is scattered in the manuscript, but a clear campaign section shortly discussing the geophysical sites, number of flights, time and duration of flights, SZA change during flights, environmental conditions, e.g. cloud fraction, etc. would improve interpretation. Information on the exact dates are for example provided for the first time on p. 5, L.9 in the context of a discussion on BRDF derived albedo.

-Section 2.1 is unbalanced when compared to 2.2 and 2.3 and not structured well. Maybe make a separation between instrument and retrieval description. I also suggest to move parts to campaign section (e.g. P.4,L.13 to L.18).

-Discussion of rasters on p. 5, L.33 should better be moved to another section (new sub-section under section 3 "Results") as it shows actual results, PANDORA locations, etc. Moreover some details are lacking again, e.g. "morning flight", is this at 6 AM or 10 AM or...? This is important for interpretation of for example traffic plumes.

-Introduction: please provide a more detailed overview of currently existing UV-VIS mapping instruments for completeness of the literature overview.

P.2, L.35: Scanning at different azimuth angles can improve the representativeness of the ground-based data when compared to satellite retrievals. Even though the focus is on direct-sun observations in this work, the potential of multi-azimuth scanning to cope with the representativeness problem should be discussed in the introduction and/or conclusion. Especially as you mention "Best practices for satellite validation strategies" in the abstract.

P.4, L.3: Please explain why this small window for $NO_2$ retrievals is selected and not a window which is better comparable with OMI, TROPOMI $NO_2$ retrievals. Moreover, in Nowlan et al. (2016) the fitting window 420-465 nm was used. Please properly refer to where the DOAS retrieval settings can be found or provide them in the manuscript.

It would be interesting for Section 3.2 to differentiate between PANDORA stations with a heterogeneous and rather homogeneous $NO_2$ distribution around the station (semi-background stations). This could be done based on the spatial distribution around the station observed by GEOTASO. The impact of the mismatched spatial representation, as reported in section 3.2, is expected to decrease in case of a more homogeneous distribution. An effort in this direction is done by differentiating based on the magnitude of PANDORA TropVCs and assuming that high $NO_2$ values can be associated with localized features and thus strong heterogeneity (P.17, L.11). This is true, but low PANDORA TropVCs do not necessarily mean that the $NO_2$ field is semi-background and that there are no fine-scale plumes around the site. It could be depending on the viewing geometry of the direct-sun measurement which is missing a plume or plumes present around the station.

Conclusion: Related to the mismatched spatial representation reported in 3.2, please provide as well suggestions to solve this for future operational validation of satellite data such as TROPOMI and TEMPO, based on ground-based stations. As airborne data is collected on campaign basis, it will not always be available. Do you consider more viewing angles (thus not only direct-sun observations) to add as an additional constraint? Maybe using AQ model data around the stations, providing knowledge on the emission sources and direction of the plumes? This could help to assess if the viewing direction is hitting a (localized) plume or not.

**Minor comments**

P.2, L.44: Replace "the" by "a" unique perspective. Mobile-DOAS measurements for example can provide as well a unique insight in the spatial variability around a ground-based station.

P.4, L.9: The overlapping GEOTASO retrievals also allow an interesting way to compare coinciding VCDs and assess the GEOTASO product quality (even if we know that the NO2 field is changing). Has this been done?

P.4, L.25: I have a bit hard time to interpret this. You are discussing larger retrieval uncertainty due to less signal (low albedo, large SZA)? The multi-linear regression is applied on which data? Details are lacking on time of flights (or SZA) to properly interpret this. Moreover, I assume flights took place with SZA smaller than 60°, so the uncertainty related to SZA should be smaller as reported?

P.5, L.26: "simplified from.." → not sure if it is needed to cite this reference as it is a commonly used equation and appears in earlier publications.

P.5, L.30: Please provide some more info on the reference used. Do you average spectra over a certain period to reduce noise? Do you use a different reference per spatial pixel in order to reduce striping effects? Is the instrument stable enough to use a single reference for the whole campaign period?

P.15, L.35: To improve readability, please repeat again explicitly which exact data sets (which days) you are comparing here.

P.15, L.30: Could you see any consistency with traffic peak times (or diurnal photochemistry) when looking at TropVC values in the 4 grids acquired on the same day?

P.15, L.47: True for ground-based vs satellite retrievals. Maybe highlight here again the advantage of airborne measurements, able to fully cover satellite pixels at high resolution.

Figure 9 and 11: Please provide the fit parameters and correlation as well in the plot or legend.

Figure 11: Pandora min and max data during the overpass +- 5 minutes can sometimes show large variations and maybe too large to be fully attributed to temporal variations. Can you shortly explain this? Outliers? I assume you lack good statistics to use 2 x st.dev. or 10-90 percentile for the whiskers.

In general for Section 3.3: Please compare your results as well with other studies that have done efforts to compare OMI with ground-based measurements, e.g. with MAX-DOAS and assess if your findings are consistent.

Section 3.3: Did you make use of the OMI averaging kernels to smooth the PANDORA VCDs in order to take into account differences in sensitivity?

**Technical corrections**

P.5, L.25: Remove "the"

P.7, L.36: Formulation is confusing. Maybe mention that these are the DSCD precision and accuracy or provide a typical value for the AMF, e.g. "assuming an AMF…."

P.18, L.29: Please change "city-to-regional spatial scale" to "regional spatial scale"

P.18, L.31: Please remove "very". A priori profiles and surface reflectances can be retrieved at much higher resolutions.

---

## Referee Comment (RC2) · Anonymous Referee #2 · 8 Aug 2019

Review of Judd et al. -- Evaluating the impact of spatial resolution on tropospheric NO2 column comparisons within urban areas using high-resolution airborne data

The authors investigate the impact of spatial variability on correlative studies for the validation of satellite trace gas products with ground-based instruments. High resolution airborne imaging DOAS measurements from GeoTASO, ground-based Pandora, and two OMI satellite products are used.

The paper is generally well written and of significance for the validation of satellite trace gas retrievals. I therefore recommend publication in AMT after some minor revisions.

**General comments**

- The introduction should contain an overview of existing airborne imaging DOAS systems
- Information about the campaign is scattered in the manuscript. A solution could be a campaign sections, with a description of the target sites (urban/rural, # of inhabitants, industrial emitters…), as well as a description of the measurement conditions (Date, time of day, SZA, AOD, meteorology…) maybe as a table…
  Here you should also give an overview of the flights presented in this study to help the reader
- I could not find any information about the DOAS fit settings used (except the fit window). Please provide that information (cross-sections, I0, Ring, …), e.g. in a table.
- You often refer to differences in spatial resolution of the a priori inputs. It would be nice if you could provide the spatial (and temporal?) resolution of the SP and BEHR products.
- You often state that the Pandora Pandora measurements are representative up to a certain pollution scale. In my opinion this statement is not correct. The representativeness depends on the spatio-temporal variability of NO2 at the Pandora location. You use an NO2 threshold to filter out data with large variability, but the magnitude of NO2 itself is not an indicator for the representativeness. I think you should amend the manuscript to reflect the differences between the physical reasons (variability) and the methodology (filtering by threshold).

**Detailed comments & technical corrections:**

| Page | Line | Comment |
|---|---|---|
| 2 | 26ff | You mention: "development of […] instruments" but you then only write about GeoTASO. I think you also had GCAS in mind. I suggest to explicitly mention it. Here you could also refer to other instrument previously used. |
| 2 | 35 | "… such as NO2." NO2 is not a product, but a chemical species. Suggestion: NO2 tropospheric vertical column densities. |
| 3 | 21 | What is the field of view in degrees? |
| 4 | 3 | Is the spectral resolution constant over the spatial dimension? If not, how does it vary? |
| 4 | 7 | How many spectra are co-added for the 250m (or what is the speed of the aircraft) |

| | | |
|---|---|---|
| 4 | 13-18 | I think this paragraph could be moved to the 'campaign' section suggested in the general comments |
| 4 | 42 | Are the inputs for the RT simulations generalized assumptions or do you perform specific calculations for each flight? |
| 5 | 31 | Do you account for diurnal changes in the stratospheric column, or do you assume a fixed value per campaign site? |
| 7 | 5 | "subtle influence of a varying $NO_2$ shape factor is visible in the AMF", I assume you are referring to the rectangular pattern above the lake. Do you consider a change of ~50% to be subtle? Is this pattern coming from the CMAQ model grid boxes? Could it be that these patterns are caused by averaging of flights performed under varying geometries? It would be nice to also see the flight tracks. Maybe you can add them as lines in Figure 1. |
| 7 | 40 | Why do you use DU now? Please also write the molec / cm2, e.g. 0.05DU (1.34e15 molec / cm2) |
| 8 | 13 | What is the resolution of the SP and BEHR a priori data inputs? |
| 12 | 17-20 | For a larger AMF the a priori profile must be shifted towards higher altitude. Or in other words, the $NO_2$ is in higher altitudes than assumed by the model. Is an uplift of $NO_2$ likely in a sea breeze front? Please briefly explain the mechanism for an uplift of $NO_2$. |
| 12 | 30 | There is a small hill ~200-300 m, in the area around "CalTech" and "LA main street". Could it be that there there are issues related to the surface air pressure in the RT simulations? No need for a detailed discussion, but you should have a look at the pressure profiles for this area. |
| 13 | F7 | Please also add the date as a title in the figure as you did in figures before |
| 14 | F8 | Please also add the date as a title in the figure as you did in figures before |
| 15 | 6 | "during these rasters": I think "rasters" should be replaced by flights / flight patterns or similar. |
| 15 | 30 | Do you have an explanation why the magnitude of $NO_2$ levels is so different between L.A. / Chicago; or also between the different days. Is that related to wind speed? You should provide some information, see general comments. |
| 15 | 34 | How exactly do you do the coincidence analysis? In the co-added rasters you cannot take the median in a 750m radius. What is the GeoTASO time – the average time? |
| 16 | 1 | "solely" is a strong word. I am sure there are other minor reasons for mismatches. Maybe better use "driven by", "mainly caused by" or similar. |
| 17 | 14 | I do not agree, that the representativeness of Pandora measurements depends on the pollution scale, such as a threshold value. As you write in line 9ff. the Pandora measurements are representative in areas with small sub-satellite-pixel variability. It is true that small variability is usually found at stations in background (non-urban) regions with low pollution levels. Filtering by pollution levels is a very basic and simple approach to the problem of spatial representativeness. Though applying refined approaches it may be out of scope for this study, I think you should provide ideas how to improve this idea. |
| 19 | 22 | Do you have an explanation why the correlation decreases for large pixel sizes and low resolution a priori data? |
| 20 | 32 | "pollution scale appears to be…" for the investigated times and areas |
| 20 | 34 | Correct line break in unit |

---

## Author Comment (AC1) · 6 Sep 2019

The comment was uploaded in the form of a supplement:
https://www.atmos-meas-tech-discuss.net/amt-2019-161/amt-2019-161-AC1-supplement.pdf

---

## Author Response (AR1)

*We would like to thank both reviewers for the helpful suggestions and questions and have made changes to the manuscript to reflect the suggestions made. Individual comments from the reviewer are bolded below with our response in italics. The comments and responses are followed by the revised manuscript with the tracked changes highlighted.*

*Reviewer 1:*

**The paper studies the impact of typical (relatively coarse) spatial resolutions of past, present and planned satellite missions on tropospheric NO₂ retrievals over areas characterized by a strong spatiotemporal variability in the NO₂ field. High resolution airborne GEOTASO NO₂ VCDs as well as TEMPO, TROPOMI and OMI VCDs, simulated from the GEOTASO observations, are compared with coincident observations at 10 PANDORA sites at the western shore of Lake Michigan and over the Los Angeles Basin. By combination of the different data sets, the work provides an interesting insight into the spatiotemporal tropospheric NO₂ variability over polluted areas. The impact of mismatched spatial representation has been quantified. The results and discussions are valuable in the assessment of validation strategies for the future generation of air pollution satellites. The scientific content of the paper fits well within the scope of AMT, and the manuscript is well-written and generally well-structured. However, some revisions (detailed below) should need to be conducted in the paper before publication.**

**General comments**

**The first two sections are missing some essential information that are valuable to the reader to better interpret the results and the geophysical parameters. I suggest the following:**

**-Please add a "Campaign" section. Some information about the campaigns is scattered in the manuscript, but a clear campaign section shortly discussing the geophysical sites, number of flights, time and duration of flights, SZA change during flights, environmental conditions, e.g. cloud fraction, etc. would improve interpretation. Information on the exact dates are for example provided for the first time on p. 5, L.9 in the context of a discussion on BRDF derived albedo.**

*We added a campaign section to isolate information about these campaigns and the details about the measurements taken in each region. We also added a table that summarizes details from each flight in terms of SZA, areas mapped, pollution scales, and cloud conditions.*

**-Section 2.1 is unbalanced when compared to 2.2 and 2.3 and not structured well. Maybe make a separation between instrument and retrieval description. I also suggest to move parts to campaign section (e.g. P.4,L.13 to L.18).**

*We moved the relevant information to the campaign section and reorganized Section 2 through the use of additional headers for separating subsections. We agree that the sections between OMI, Pandora, and GeoTASO are not equally weighted, but did not see a way to shorten the GeoTASO section as the retrieval is not yet standard whereas the Pandora and OMI products are more standardized and better characterized through the literature.*

*However, we moved the GeoTASO section after the Pandora and OMI and campaign sections, which we think helps address this comment by providing a smoother transition between the retrieval descriptions and the results.*

**-Discussion of rasters on p. 5, L.33 should better be moved to another section (new sub-section under section 3 "Results") as it shows actual results, PANDORA locations, etc. Moreover some details are lacking again, e.g. "morning flight", is this at 6 AM or 10 AM or…? This is important for interpretation of for example traffic plumes.**
*This figure and text aid discussion about the retrieval sensitivity and uncertainty, which we feel is needed in the data section. This makes the data section longer, which isn't ideal, but through the reorganization of section 2 (see previous comment) we think the intent of the comment has been addressed by improving the flow. We also added times to the captions of each of the rasters. Both flights are representative of mid-morning in their respective local times.*

**-Introduction: please provide a more detailed overview of currently existing UV-VIS mapping instruments for completeness of the literature overview.**
*We added a brief literature review of recent work using other mapping spectrometers in the introduction.*

**P.2, L.35: Scanning at different azimuth angles can improve the representativeness of the ground-based data when compared to satellite retrievals. Even though the focus is on direct-sun observations in this work, the potential of multi-azimuth scanning to cope with the representativeness problem should be discussed in the introduction and/or conclusion. Especially as you mention "Best practices for satellite validation strategies" in the abstract.**

*We agree that scanning at different azimuth angles adds a unique dimension to a future validation studies and should be explored in future work. We have added some discussion in section 3.3 in the context of a literature review of MAX-DOAS comparisons to OMI. Because the direct-sun $NO_2$ retrieval from Pandora is what is currently available as a standard product, we emphasize in the revised manuscript that these conclusions are drawn from comparisons to direct-sun retrievals.*

**P.4, L.3: Please explain why this small window for NO2 retrievals is selected and not a window which is better comparable with OMI, TROPOMI NO2 retrievals. Moreover, in Nowlan et al. (2016) the fitting window 420-465 nm was used. Please properly refer to where the DOAS retrieval settings can be found or provide them in the manuscript.**
*We apologize as there was a typographical error in the manuscript. The DOAS window spans from 425-460 nm, not 435-460 nm. We fixed this in this revision. This window is similar to Nowlan et al. (2016) and the same as Lamsal et al. (2017). We also added more details on the DOAS retrieval within section 2.4.1 in the revised manuscript.*

**It would be interesting for Section 3.2 to differentiate between PANDORA stations with a heterogeneous and rather homogeneous NO2 distribution around the station (semi-background stations). This could be done based on the spatial distribution around the station observed by GEOTASO. The impact of the mismatched spatial representation, as reported in section 3.2, is expected to decrease in case of a more homogeneous distribution. An effort in this direction is done by differentiating based on the magnitude of PANDORA TropVCs and assuming that high NO2 values can be associated with localized features and thus strong heterogeneity (P.17, L.11). This is true, but low PANDORA TropVCs do not necessarily mean that the NO2 field is semi-background and that there are no fine-scale plumes around the site. It could be depending on the viewing geometry of the direct-sun measurement which is missing a plume or plumes present around the station.**

*You are correct that low TropVCs do not necessarily reflect a homogeneous environment. The difference between individual coincident GeoTASO and Pandora measurements may be larger than the uncertainty of both measurements due to spatial representation mismatches. In fact, none of the Pandora sites are 100% homogeneous in all cases based on looking at the subpixel variability around the sites from the airborne spectrometer during these campaigns. Thank you for this comment as it sparked additional discussion of this section as well as the conclusions. Please see the revised manuscript for these changes.*

**Conclusion: Related to the mismatched spatial representation reported in 3.2, please provide as well suggestions to solve this for future operational validation of satellite data such as TROPOMI and TEMPO, based on ground-based stations. As airborne data is collected on campaign basis, it will not always be available. Do you consider more viewing angles (thus not only direct-sun observations) to add as an additional constraint? Maybe using AQ model data around the stations, providing knowledge on the emission sources and direction of the plumes? This could help to assess if the viewing direction is hitting a (localized) plume or not.**

*We did not consider more viewing angles as we do not have measurements from Pandora to consider this from these campaigns and therefore cannot make conclusions about how this information would add to the results. However, the focus here is on the now-standard Pandora direct-sun measurements that can be accessed in near-real time for any operating instrument. Overall, these results show that at a TROPOMI spatial scale or finer, direct-sun observations are highly correlated with satellite-like products even in polluted environments (that can still be subject to variability).*

*To specifically address the comment about adding recommendations for future validation work with direct-sun columns from Pandora, we have added additional suggestions to the final paragraph of the Conclusions section. Please see the revised manuscript for these changes.*

**Minor comments**
**P.2, L.44: Replace "the" by "a" unique perspective. Mobile-DOAS measurements for example can provide as well a unique insight in the spatial variability around a ground-based station.**

*Change made as suggested.*

**P.4, L.9: The overlapping GEOTASO retrievals also allow an interesting way to compare coinciding VCDs and assess the GEOTASO product quality (even if we know that the NO2 field is changing). Has this been done?**
*There are a few pixels that overlap between adjacent flight lines, and we have done some preliminary analysis but have not yet compared the overlapping pixels quantitatively. It does not appear to be a large effect in the data reported here. It could be an interesting study in the future as an interesting way to evaluate the impact of BRDF on the retrieval.*

**P.4, L.25: I have a bit hard time to interpret this. You are discussing larger retrieval uncertainty due to less signal (low albedo, large SZA)? The multi-linear regression is applied on which data? Details are lacking on time of flights (or SZA) to properly interpret this. Moreover, I assume flights took place with SZA smaller than 60°, so the uncertainty related to SZA should be smaller as reported?**
*We reworded this paragraph to clarify to which data the statistics are applied. Additional data about the flight times/SZAs have also been added in the campaign section for more interpretation.*

**P.5, L.26: "simplified from.."    not sure if it is needed to cite this reference as it is a commonly used equation and appears in earlier publications.**
*Removed (simplified from……)*

**P.5, L.30: Please provide some more info on the reference used. Do you average spectra over a certain period to reduce noise? Do you use a different reference per spatial pixel in order to reduce striping effects? Is the instrument stable enough to use a single reference for the whole campaign period?**

*We added information (dates/areas/sza/time) on the references and noted that these were clean/homogeneous regions. You are correct that we use a difference reference for each spatial pixel to avoid striping. The instrument was stable throughout the 2017 flights and no changes were made to the instrument during the campaign, which allowed us to use just one reference for each region and therefore reduce bias associated with having a suite of different references.*

**P.15, L.35: To improve readability, please repeat again explicitly which exact data sets (which days) you are comparing here.**
*We reworded the opening sentences to that paragraph to clarify which datasets are used and how they are compared to the upscaled version.*

**P.15, L.30: Could you see any consistency with traffic peak times (or diurnal photochemistry) when looking at TropVC values in the 4 grids acquired on the same day?**
*There is a diurnal pattern that occurs with the 4 rasters on the same day in this case. We saw the peak in $NO_2$ early-mid morning and it decreased through the day. This seems to match*

*traffic patterns in the area, but in-depth analysis is pending updated emission inventories being developed for the Chicago region. We have attempted to clarify the wording of this sentence.*

**P.15, L.47: True for ground-based vs satellite retrievals. Maybe highlight here again the advantage of airborne measurements, able to fully cover satellite pixels at high resolution.**
*We added text to highlight the advantage of the airborne measurements in these types of experiments.*

**Figure 9 and 11: Please provide the fit parameters and correlation as well in the plot or legend.**
*We added the statistics to Figure 11, however felt they could not be legibly added to Figure 9 so instead added to the Figure 9 capture a referral to Table 3, which presents the statistics.*

**Figure 11: Pandora min and max data during the overpass +- 5 minutes can sometimes show large variations and maybe too large to be fully attributed to temporal variations. Can you shortly explain this? Outliers? I assume you lack good statistics to use 2 x st.dev. or 10-90 percentile for the whiskers.**
*Your assumption is correct. In a 10-minute window, there is a maximum of 7 measurements from Pandora, which is not sufficient for reporting statistics other than max/min. We have explored these max/min values at times when they have a very large range in this short time window, and the variation at these sites (most often the Schiller Park location) appears to be real. We have not found individual outliers suggestive of some sort of instrument or retrieval error, but rather large swings in the magnitude over multiple measurements. A couple examples of this are shown in Figures 4 and 6.*

**In general for Section 3.3: Please compare your results as well with other studies that have done efforts to compare OMI with ground-based measurements, e.g. with MAX-DOAS and assess if your findings are consistent.**
*We extended the discussion in Section 3.3 to include previous studies comparing measurements with OMI. Please refer to the revised text within section 3.3.*

**Section 3.3: Did you make use of the OMI averaging kernels to smooth the PANDORA VCDs in order to take into account differences in sensitivity?**
*We did not. As the Pandora VCDs are not dependent on their own a priori assumptions (equally sensitive vertically and AMF is geometric), this is not necessary as it would be when comparing to measurements requiring their own a priori and sensitivity profiles (i.e., MAX-DOAS or model comparisons).*

**Technical corrections**
**P.5, L.25: Remove "the"**
**P.7, L.36: Formulation is confusing. Maybe mention that these are the DSCD precision and accuracy or provide a typical value for the AMF, e.g. "assuming an AMF…."**
**P.18, L.29: Please change "city-to-regional spatial scale" to "regional spatial scale"**

**P.18, L.31: Please remove "very". A priori profiles and surface reflectances can be retrieved at much higher resolutions.**

*All technical corrections were made as suggested.*

*Reviewer 2:*

**Review of Judd et al. -- Evaluating the impact of spatial resolution on tropospheric NO2 column comparisons within urban areas using high-resolution airborne data**

**The authors investigate the impact of spatial variability on correlative studies for the validation of satellite trace gas products with ground-based instruments. High resolution airborne imaging DOAS measurements from GeoTASO, ground-based Pandora, and two OMI satellite products are used. The paper is generally well written and of significance for the validation of satellite trace gas retrievals. I therefore recommend publication in AMT after some minor revisions.**

**General comments**

**The introduction should contain an overview of existing airborne imaging DOAS systems**

*We have added a short literature review of other mapping spectrometers in the introduction.*

**Information about the campaign is scattered in the manuscript. A solution could be a campaign sections, with a description of the target sites (urban/rural, # of inhabitants, industrial emitters…), as well as a description of the measurement conditions (Date, time of day, SZA, AOD, meteorology…) maybe as a table… Here you should also give an overview of the flights presented in this study to help the reader**

*We have added a campaign section at the beginning of section 2. This includes an overview of the purpose of each campaign, number of flights, total hours, areas measured as well as some discussion on typical meteorology and SZA and a table that summarizes some details about each flight.*

**I could not find any information about the DOAS fit settings used (except the fit window). Please provide that information (cross-sections, I0, Ring, …), e.g. in a table.**

*We have added information about the DOAS fit under the Airborne $NO_2$ retrieval section (now Section 2.4.1).*

**You often refer to differences in spatial resolution of the a priori inputs. It would be nice if you could provide the spatial (and temporal?) resolution of the SP and BEHR products.**

*The spatial resolution of the a priori input to the AMF calculation is finer for BEHR than SP. Discussion about the specific differences on surface reflectivity, terrain pressure, and NO2 profile are added to the OMI section.*

**You often state that the Pandora measurements are representative up to a certain pollution scale. In my opinion this statement is not correct. The representativeness depends on the spatio-temporal variability of NO2 at the Pandora location. You use an NO2 threshold to filter out data with large variability, but the magnitude of NO2 itself is not an indicator for the representativeness. I think you**

**should amend the manuscript to reflect the differences between the physical reasons (variability) and the methodology (filtering by threshold).**

*We agree with this statement and in hindsight believe the original statement was worded poorly (and backwards, as it is the spatial scale of the satellite retrieval that may not be representative of the features Pandora is observing). We amended the text to reflect these points and attempted to more clearly express the purpose of applying the thresholds.  Please see the appropriate areas of the abstract, Sect 3.2, and conclusions in the revised submission to see how these points were edited to emphasize these conclusions.*

| Detailed comments & technical corrections: Page | Line | Comment |
| --- | --- | --- |
| 2 | 26ff | **You mention: "development of […] instruments" but you then only write about GeoTASO. I think you also had GCAS in mind. I suggest to explicitly mention it. Here you could also refer to other instrument previously used.** *Added GCAS as well since this is also a NASA supported instrument.  There is also now a short literature review on other airborne spectrometer research efforts as well.* |
| 2 | 35 | **"… such as NO2." NO2 is not a product, but a chemical species. Suggestion: NO2 tropospheric vertical column densities.** *Added tropospheric vertical column densities as suggested.* |
| 3 | 21 | **What is the field of view in degrees?** *It is 45 degrees, which is stated within the section.* |
| 4 | 3 | **Is the spectral resolution constant over the spatial dimension? If not, how does it vary?** *The spectral resolution does not vary over the spatial dimension (within 0.01nm).* |
| 4 | 7 | **How many spectra are co-added for the 250m (or what is the speed of the aircraft)** *The across track dimension is separated into 33 across track positions with ~30 images per bin. For the along track dimension, the code considers the median aircraft speed at altitude (typically about 100 m/s) and then calculates how many along track spectra are needed to get closest to 250m.  Typically, it comes results in around 300 spectra coadded to get 250 x 250 m.* |
* * *

[revised manuscript text omitted]

**Page 10: [1] Deleted  Judd, Laura M. (LARC-E303)[UNIVERSITIES SPACE RESEARCH ASSOCIATION]**        **9/13/19 12:49:00 PM**